# Tunable positions of Weyl nodes via magnetism and pressure in the ferromagnetic Weyl semimetal CeAlSi

Erjian Cheng [1,2] ✉, Limin Yan[3,4], Xianbiao Shi[5,6], Rui Lou [1,7,8] ✉,
Alexander Fedorov [1,7,8], Mahdi Behnami[1], Jian Yuan[9], Pengtao Yang [10,11],
Bosen Wang [10,11], Jin-Guang Cheng [10,11], Yuanji Xu[12], Yang Xu [13], Wei Xia[9],
Nikolai Pavlovskii[14], Darren C. Peets [14], Weiwei Zhao [5,6], Yimin Wan[15],
Ulrich Burkhardt [2], Yanfeng Guo [9], Shiyan Li [15,16,17], Claudia Felser[2],
Wenge Yang [3] ✉ & Bernd Büchner [1,18] ✉

The noncentrosymmetric ferromagnetic Weyl semimetal CeAlSi with simultaneous space-inversion and time-reversal symmetry breaking provides a unique platform for exploring novel topological states. Here, by employing multiple experimental techniques, we demonstrate that ferromagnetism and pressure can serve as efficient parameters to tune the positions of Weyl nodes in CeAlSi. At ambient pressure, a magnetism-facilitated anomalous Hall/Nernst effect (AHE/ANE) is uncovered. Angle-resolved photoemission spectroscopy (ARPES) measurements demonstrated that the Weyl nodes with opposite chirality are moving away from each other upon entering the ferromagnetic phase. Under pressure, by tracing the pressure evolution of AHE and band structure, we demonstrate that pressure could also serve as a pivotal knob to tune the positions of Weyl nodes. Moreover, multiple pressure-induced phase transitions are also revealed. These findings indicate that CeAlSi provides a unique and tunable platform for exploring exotic topological physics and electron correlations, as well as catering to potential applications, such as spintronics.

Over the last decade, the advancement of theoretical predictions and experimental validations in topological semimetals has hastened the progress of research on topological states of matter and topotronics[1–4]. In topological materials, particularly significant efforts have been devoted to searching for and characterizing novel topological states due to their exotic properties, such as the presence of low-energy excitations, extremely large and unsaturated magnetoresistance (MR), topological surface states, Fermi arcs, chiral anomaly[1–4]. The addition of magnetic elements in magnetic topological materials (MTMs) breaks the time-reversal (TR) symmetry, leading to various intriguing phenomena[5–9], such as intrinsic

anomalous Hall/Nernst[1–4,10] and topological Hall/Nernst effects[11–17], and topological magnetic textures (for example, skyrmions[11–18], hedgehogs[19,20], merons[21], magnetic bubbles[13], hopfions[22]). The intricate relationship between magnetism and topology continues to be complex, yet holds promise for unlocking new and exotic topological states. Hitherto research on this interplay is limited to a few cases, for example, a magnetic field-induced ideal type-II Weyl state in $Mn(Bi_{1-x}Sb_x)_2Te_4$[23,24], a magnetic-exchange-induced Weyl state in $EuCd_2Sb_2$[25], a spin-fluctuation-induced Weyl semimetal state in $EuCd_2As_2$[17,26], a magnetism-induced topological transition in $EuAs_3$[27], and magnetization-tunable Weyl states in $EuB_6$[28], etc. To exploit

more novel phenomena and elaborate the relationship, more systems are called for.

Recently, the noncentrosymmetric *RAlPn* series were proposed and demonstrated to be Weyl semimetals[21,29–49], which host low-energy excitations, namely Weyl fermions described by the Weyl equation with 2 × 2 complex Pauli matrices[1–4]. Weyl fermions arise in the vicinity of a doubly degenerate electronic band crossing point (the Weyl node), and a pair of Weyl nodes possess opposite chirality[1–4]. In general, to attain Weyl states, space-inversion (SI) or TR symmetry should be broken, and there are a few cases in which SI and TR symmetries are simultaneously broken[1–4]. For the nonmagnetic LaAl*Pn*, SI symmetry is naturally broken, and the system hosts two types of Weyl states (type-I and type-II)[31,32]. Moreover, a spin Hall angle that is comparable to *M*Te$_2$ (*M* = W, Mo) has been predicted in LaAl*Pn*[50–54]. More intriguingly, pressure-induced superconductivity and robust topology against pressure up to 80.4 GPa have been uncovered in LaAl*Pn*[55]. In contrast to LaAl*Pn*, both SI and TR symmetries are broken in the magnetic siblings, i.e., *RAlPn* (*R* = Ce, Pr, Nd, Sm, *Pn* = Si, Ge), rendering them rare cases for studying novel topological properties with the simultaneous breaking of SI and TR symmetries[21,29,36,39,49,56].

CeAlSi is a ferromagnetic Weyl semimetal with noncollinear magnetic ordering, and electrical transport measurements revealed an anisotropic AHE, a loop-shaped Hall effect (LHE) in the ferromagnetic state, and a nontrivial Berry phase[41]. Moreover, it has been demonstrated that both the AHE and the ANE are evident even within the paramagnetic state. This underscores the proposal that the **k**-space topology plays a crucial role in determining transport properties at both low and high temperatures[57]. Previous ARPES experiments in the paramagnetic phase unveiled possible surface Fermi arcs and linearly dispersing conical features that correspond to the Weyl cones, further implying the existence of Weyl fermions[42]. The flat band stemming from Ce 4*f* electrons was also detected near the Fermi level (*E$_F$*), indicating that electron correlations may also play a role[42]. Scanning superconducting quantum interference device (sSQUID) and magneto-optical Kerr effect (MOKE) microscopy on CeAlSi found the presence of nontrivial chiral domain walls that contributed to the topological properties[43–45].

According to DFT calculations, the Weyl nodes in CeAlSi arise from the SI symmetry breaking, and the TR symmetry breaking with the inclusion of magnetism just shifts the positions of Weyl nodes in the Brillouin zone (BZ) as the ferromagnetism acts as a simple Zeeman coupling[30,41], which lacks experimental verification. Upon compressing CeAlSi up to ~3 GPa, it was proposed that the electronic structure and magnetic structure of CeAlSi remain nearly unchanged, but the AHE and LHE are suppressed[45]. Nevertheless, the evolution of band structure and topology under higher pressure has not been elaborated[40,45]. Since ferromagnetism is proposed to serve as an efficient parameter to tune the positions of Weyl nodes, therefore how and to what extent the Weyl nodes in CeAlSi evolve with pressure remains elusive.

In this work, by resorting to electrical and thermoelectrical transport, ARPES, high-pressure techniques, and band calculations, we systematically study the band structure and topological properties of CeAlSi. At ambient pressure, both ANE and AHE in CeAlSi are unveiled. They arise in the paramagnetic state, and then are enhanced when temperature approaches the ferromagnetic transition temperature (*T$_C$*), indicative of the interaction of magnetism and topology. The anomalous Hall conductivity ($\sigma_{xy}^A$) and the anomalous Nernst conductivity ($\alpha_{xy}^A$) follow the Mott relation, and the latter is the derivative of the former at *E$_F$*[10,58]. When *E$_F$* crosses the Weyl points, $\sigma_{xy}^A$ reaches a maximum, while $\alpha_{xy}^A$ peaks when *E$_F$* shifts away from the Weyl points[57]. Since the TR symmetry breaking due to the inclusion of ferromagnetism in CeAlSi does not change the classification and the quantity of Weyl nodes but just shifts their positions, the enhancement

of AHE/ANE may be related to the increased distance of Weyl nodes with opposite chirality. This is confirmed by our joint ARPES and DFT investigations. The magnetic tunability of the bulk and surface band structure is experimentally realized, which, to the best of our knowledge, has never been reported before in other *RAlPn* compounds. Under pressure, an enhancement and a sign change of AHE take place. Based on band calculations, we found that pressure has a similar effect on Weyl nodes as magnetism. In addition, multiple pressure-induced phase transitions are discovered, i.e., a pressure-induced Lifshitz transition at ~10 GPa, a magnetic transition from the ferromagnetic state to a paramagnetic state beyond ~20 GPa, and a structural phase transition at ~40 GPa. These results suggest that CeAlSi provides a unique and tunable platform to study novel topological states, the interplay between magnetism and topology, and topological properties with electron correlation effect.

## Results

### Anomalous magneto-transverse transport in CeAlSi

CeAlSi crystallizes in the tetragonal structure with the space group *I*4$_1$*md* (No. 109)[32,41], as shown in Fig. 1a. High-quality single crystals of CeAlSi are synthesized through a flux method[32,41]. The largest natural surface is the *ab* plane [Supplementary Fig. 2b, c]. CeAlSi possesses in-plane (the *ab* plane) noncollinear ferromagnetic ordering below ~10 K defined according to magnetization, which is also evident in resistivity [Fig. 1c][41]. Figure 1b displays the calculated electronic band structure and associated Berry curvature. Figure 1d shows the Hall resistivity of CeAlSi at different temperatures. There is a turning point at ~2.5 T, above which the Hall resistivity profile with a positive slope displays a linear dependence. The turning point in Hall resistivity persists up to ~10 K, and then broadens and shifts to higher fields. Above ~100 K, the Hall resistivity displays linear behavior. Figure 1f shows the magnetization at different temperatures. The magnetization above 15 K displays a linear dependence, evidencing that CeAlSi is in the paramagnetic state. When the temperature decreases below 15 K, the system approaches the regime of magnetic fluctuations, and nonlinear components start to contribute. To obtain the anomalous contributions in Hall resistivity, we subtract the linear background by adopting the expression, $\rho_{yx} = R_0 B + \rho_{yx}^A$ [41,45]. A unusual loop-shaped Hall effect (LHE), a hysteresis produced during the upward and downward scan of fields, is also verified in our sample [Supplementary Fig. 2e], as reported in previous studies[41,45]. The anomalous Hall resistivity at different temperatures is plotted in Fig. 1e. The anomalous Hall conductivity [$\sigma_{xy}^A = \rho_{yx}^A/(\rho_{yx}^{A\,2} + \rho_{xx}^2)$] and anomalous Hall angle [AHA $\equiv \arctan(\sigma_{xy}^A/\sigma_{xx}) \sim \sigma_{xy}^A/\sigma_{xx}$, $\sigma_{xx} = \rho_{xx}/(\rho_{xx}^2 + \rho_{yx}^2)$] shown in Supplementary Fig. 2f] are also calculated, as shown in Fig. 1g. The AHE (AHA) arises below ~100 K, and then ascends with temperature approaching the regime of magnetic fluctuations. When the system enters into the ferromagnetic state, the anomalous $\sigma_{xy}^A$ and AHA not vary much.

According to the Mott relation, the thermoelectric signals are proportional to the derivative of the conductivities at *E$_F$*[10], which also applies to the anomalous Hall conductivity and anomalous Nernst conductivity[10,57]. Therefore, thermoelectric transport is exquisitely sensitive to the band structure and the anomalous contributions near *E$_F$*. To further shed light on the anomalous transverse transport in CeAlSi, we performed thermoelectrical transport measurements. Figure 1h–j show the normalized magneto-Seebeck, the Nernst, and the normalized Nernst signals at selected temperatures, respectively. For the Nernst signal, a spinelike profile near zero field can be seen, which may come from conventional contributions as reported in Cd$_3$As$_2$[59]. However, for CeAlSi, the spinelike feature only appears when the system enters the ferromagnetic state, implying magnetism may adjust the band structure of CeAlSi. The field-independent anomalous components in Nernst are evident at high fields and low temperatures. To

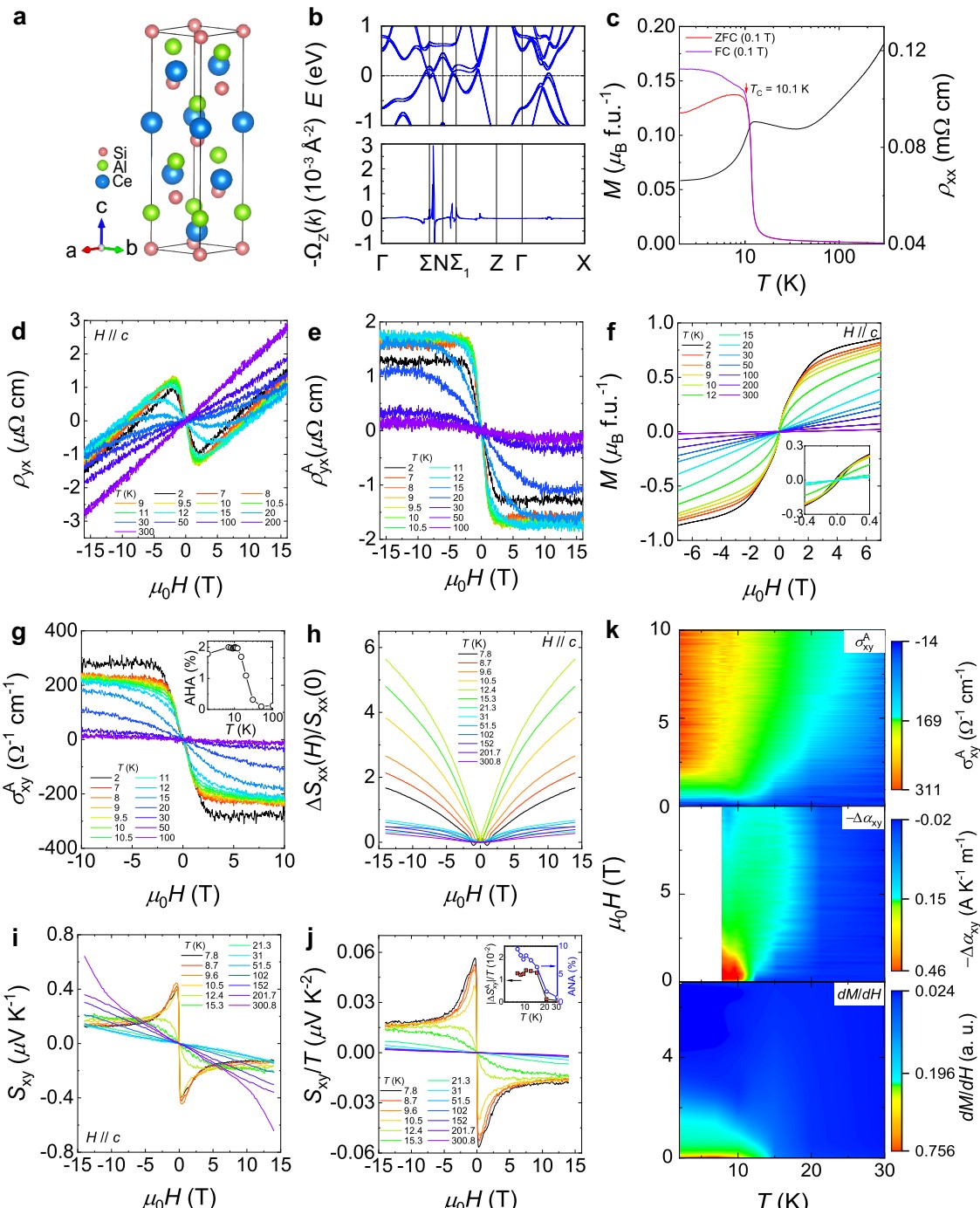

**Fig. 1 | Anomalous Hall effect (AHE) and anomalous Nernst effect (ANE) in CeAlSi. a** The schematic structure of CeAlSi with a noncentrosymmetric structure (space group of $I4_1md$). **b** Band structure and associated Berry curvature. **c** Longitudinal resistivity ($\rho_{xx}$) in zero field. Zero-field-cooling (ZFC) and field-cooling (FC) magnetization as a function of temperature for CeAlSi with the magnetic field applied along the $c$ axis. The ferromagnetic transition temperature ($T_C$) is -10.1 K, defined according to magnetization. **d** Transverse Hall resistivity ($\rho_{yx}$) different temperatures with the magnetic field applied along the $c$ axis. **e** Anomalous Hall resistivity ($\rho_{yx}^A$) at various temperatures. **f** Field dependence of magnetization at various temperatures with the magnetic field applied along the $c$ axis. Inset shows the low-field data. **g** Anomalous Hall conductivity ($\sigma_{xy}^A$) at various temperatures.

Inset displays the anomalous Hall angle (AHA). **h** Normalized magneto-Seebeck signal at different temperatures with the magnetic field applied along the $c$ axis. We plot $\Delta S_{xx}(H)/S_{xx}(0)$ [$\Delta S_{xx}(H) = S_{xx}(H) - S_{xx}(0)$] for better comparison. **i** Nernst signal at different temperatures. **j** Nernst signal normalized to the temperature at different temperatures. Inset shows the temperature dependence of the amplitude of anomalous Nernst signal normalized to the temperature ($|S_{xy}^A|/T$), and the anomalous Nernst angle (ANA). Error bars are deduced from the fit. **k** Contour plots of the $\sigma_{xy}^A$, $-\Delta\alpha_{xy}$ (see Supplementary Note 3 for more details) and the derivative of magnetization ($dM/dH$). The background color represents the magnitude of their values.

further analyze the ANE, an empirical approach is adopted[59]:

$$S_{xy} = S_{xy}^{N} + S_{xy}^{A}, \qquad (1)$$

$$S_{xy}^{N} = S_0^{N} \frac{\mu B}{1 + (\mu B)^2}, \qquad (2)$$

$$S_{xy}^{A} = \Delta S_{xy}^{A} \tanh(B/B_0), \qquad (3)$$

Here, $S_{xy}^{N}$ and $S_{xy}^{A}$ represent conventional and anomalous contributions, respectively. $S_0^{N}$, $\Delta S_{xy}^{A}$, $\mu$, and $B_0$ denote the amplitude of the conventional semiclassical contribution, the amplitude of the anomalous contribution, carrier mobility, and the saturation field above which a plateau appears. From the fit [Supplementary Fig. 3a], the amplitude of the anomalous Nernst signal ($|\Delta S_{xy}^{A}|/T$) is extracted for low temperatures, as shown in the inset of Fig. 1j. Upon decreasing temperature below ~31 K, the ANE appears and attains a plateau below 15.3 K. There is an abrupt enhancement of the anomalous Nernst angle [ANA $\equiv \arctan(\Delta S_{xy}^{A}/S_{xx}) \sim \Delta S_{xy}^{A}/S_{xx}$] when the system enters into the regime of magnetic fluctuations, and then the ANA reaches ~9.5% at 7.8 K. The AHE and ANE have also been theoretically estimated, which is nearly consistent with the experiments [Supplementary Fig. 3d, e]. To further shed light on the intrinsic AHE for CeAlSi, the anomalous Hall conductivity as a function of the longitudinal conductivity is summarized in Supplementary Fig. 5a. For the intrinsic AHE, the anomalous Hall conductivity is independent of the longitudinal conductivity ($|\sigma_{xy}^{A}|$ vs. $\sigma_{xx}$ -constant)[10,60]. Clearly, the data adhere to the universal law, verifying the intrinsic nature of the AHE in CeAlSi. We also plot the $\rho_{yx}^{A}$ as a function of $\rho_{xx}^2$ to further analyze the intrinsic AHE [Supplementary Fig. 5b].

The anomalous Nernst conductivity $-\Delta\alpha_{xy}$ is also calculated, which shows similar behavior, as displayed in Supplementary Fig. 3b. We obtain the contour plots of $\sigma_{xy}^{A}$, $-\Delta\alpha_{xy}$ and $dM/dH$ in Fig. 1k. As mentioned above, magnetization displays a nonlinear-dependence at low temperatures, and this is more evident in the $dM/dH$ plot. Above 15 K, the magnetization is linear, while $\sigma_{xy}^{A}$ and $-\Delta\alpha_{xy}$ already arise, which means that the AHE and ANE do not scale with the magnetization, and they root in topology rather than magnetism. When the system is in the vicinity of the temperature where magnetic fluctuations start to play a role, $\sigma_{xy}^{A}$ and $-\Delta\alpha_{xy}$ are significantly enhanced, implying magnetism interacts with topology, and the interplay between them facilitates the anomalous magneto-transverse transport in CeAlSi. To shed light on the anisotropic nature, we conducted the electrical and thermoelectrical measurements with in-plane magnetic field [see Supplementary Note 4 for more details]. The in-plane transport behavior is more complex due to magnetocrystalline anisotropy, but a similar conclusion can be drawn. In MTMs, the coupling between magnetic configuration and external magnetic field could produce various intermediate magnetic or topological states, and hence the variation of AHE may come from these states[23–28]. However, for CeAlSi, it was proposed that the angle between the noncollinear spins does not change with the applied magnetic field up to 8 T[41]. Therefore, the enhancement of anomalous transverse transport in CeAlSi is supposed to arise from the shift of the positions of Weyl nodes rather than intermediate states.

## Effect of magnetism on the electronic structure of CeAlSi revealed by ARPES

In order to directly uncover the intricate effect of ferromagnetism on the electronic properties of CeAlSi, we now conduct high-resolution ARPES measurements. In Fig. 2a, the $f$-electron behavior contributed by Ce is illustrated by the resonant ARPES measurements at the $N$ edge of the Ce element. The resonant enhancement at ~−0.30 eV, corresponding to the Ce $4f^{1}_{7/2}$ final state[61], is revealed at the Ce $4d \rightarrow 4f$

resonant photon energy of 121 eV. The core-level photoemission spectra in Fig. 2b show the characteristic peaks of Al-2$p$ and Si-2$p$ orbitals, where the coexistence of main peaks and shoulders is similar to the case in PrAlSi and SmAlSi[62]. Figure 2c presents the bulk and (001)-projected surface BZs with high-symmetry points of the $R$AlP$n$ family. It has been found that the ARPES intensity suffers from the large $k_z$-broadening effect in this series of compounds reflected in the observation of similar band structure in a wide vacuum-ultraviolet photon energy range[38,42,62]. The ARPES spectra would reflect the electronic states integrated over a certain $k_z$ region of the bulk BZ, therefore we use the projected 2D BZ ($\bar{\Gamma}$, $\bar{X}$, $\bar{M}$) hereafter. As shown in Fig. 2d, the overall Fermi surface (FS) topology of CeAlSi shares many similarities with those of the magnetic $R$AlP$n$ compounds like PrAlSi[62], SmAlSi[39,62], and PrAlGe[38], including the inner ($\alpha$) and outer squarelike pockets around $\bar{\Gamma}$, the dumbbell-like pockets ($\beta$) around $\bar{X}$, and the ripple-shaped FS contours across the BZ boundaries. By a closer look, we notice that there are band splittings of the $\alpha$ and $\beta$ FSs. The splitting of $\beta$ FS has been previously observed in PrAlSi and SmAlSi[62], while the split $\alpha$ FS has not been reported before in other $R$AlP$n$ materials.

To study whether the band splittings are related to ferromagnetism as well as the magnetic impact on the electronic structures of CeAlSi, we then perform temperature-dependent ARPES measurements on another sample. As shown in Fig. 2e, f, compared to the other FS contours, the evolution of the $\alpha$ and $\beta$ FSs crossing over $T_C$ is clearly revealed. In the paramagnetic phase [Fig. 2e], the splitting of $\alpha$ vanishes and the momentum splitting scale of $\beta$ increases. In Fig. 2g, h, we present the near-$E_F$ band dispersions across $T_C$ measured along cuts #a and #b [illustrated in Fig. 2d], respectively. It can be seen that the ferromagnetism has a sizeable effect on the band structure of CeAlSi, like causing the splitting of $\alpha$ band [Fig. 2g] and reducing the splitting of $\beta$ band at $E_F$ [Fig. 2h]. The behavior of $\alpha$ is compatible with the traditional Zeeman splitting[63], while the evolution of split $\beta$ could have a more complicated origin, further studies considering the correlation effects of 4$f$ electrons would be required. The marked temperature evolutions of $\alpha$ and $\beta$ bands in the energy and momentum imply the intertwined magnetism and itinerant electrons. The current observations are in sharp contrast to the temperature-independent electronic structures in ferromagnetic PrAlSi and antiferromagnetic SmAlSi, where the coupling between the spin configuration of 4$f$ states and the conduction electrons has been suggested to be negligible[62]. To our best knowledge, CeAlSi is the very rare case among the $R$AlP$n$ compounds established thus far that the experimental band structure can be successfully tuned by magnetism. Based on the previous results of magnetic $R$AlP$n$ materials[38,39,42,62], the $\alpha$ band is of surface origin, while the origin of the $\beta$ band is debated because it can be reproduced by either bulk[62] or surface[39] band calculations. With the presence of the large $k_z$-broadening effect, we deduce that the observed $\beta$ band could be contributed by both the bulk and surface features. This might be a cause of its unusual magnetic behavior. Therefore, the revealed strong interplay between the magnetism and the itinerant electrons both in bulk and on the surface suggests that, compared to other $R$AlP$n$ compounds discovered so far, CeAlSi is a very promising platform that hosting the magnetic tunability of bulk Weyl nodes and surface Fermi arcs.

To directly demonstrate the magnetism-tunable Weyl nodes in CeAlSi, we study the band dispersions composing the Weyl points above and below $T_C$. As illustrated in Fig. 2d, we examine the temperature evolutions of two pairs of Weyl points (denoted as WP1 and WP2) along cuts #c and #d, respectively. In Fig. 2i, k (cut #c), one obtains a good consistency between experiments and calculations below $E_F$. As guided by the calculations, a pair of WP1 are located at about 50 ($T > T_C$) and 40 ($T < T_C$) meV above $E_F$; it is notable that the distance of WP1 with opposite chirality greatly increases as entering the ferromagnetic phase (from ~0.006 to ~0.020 Å$^{-1}$). To validate that the WP1 moves away from each other due to the ferromagnetism, we

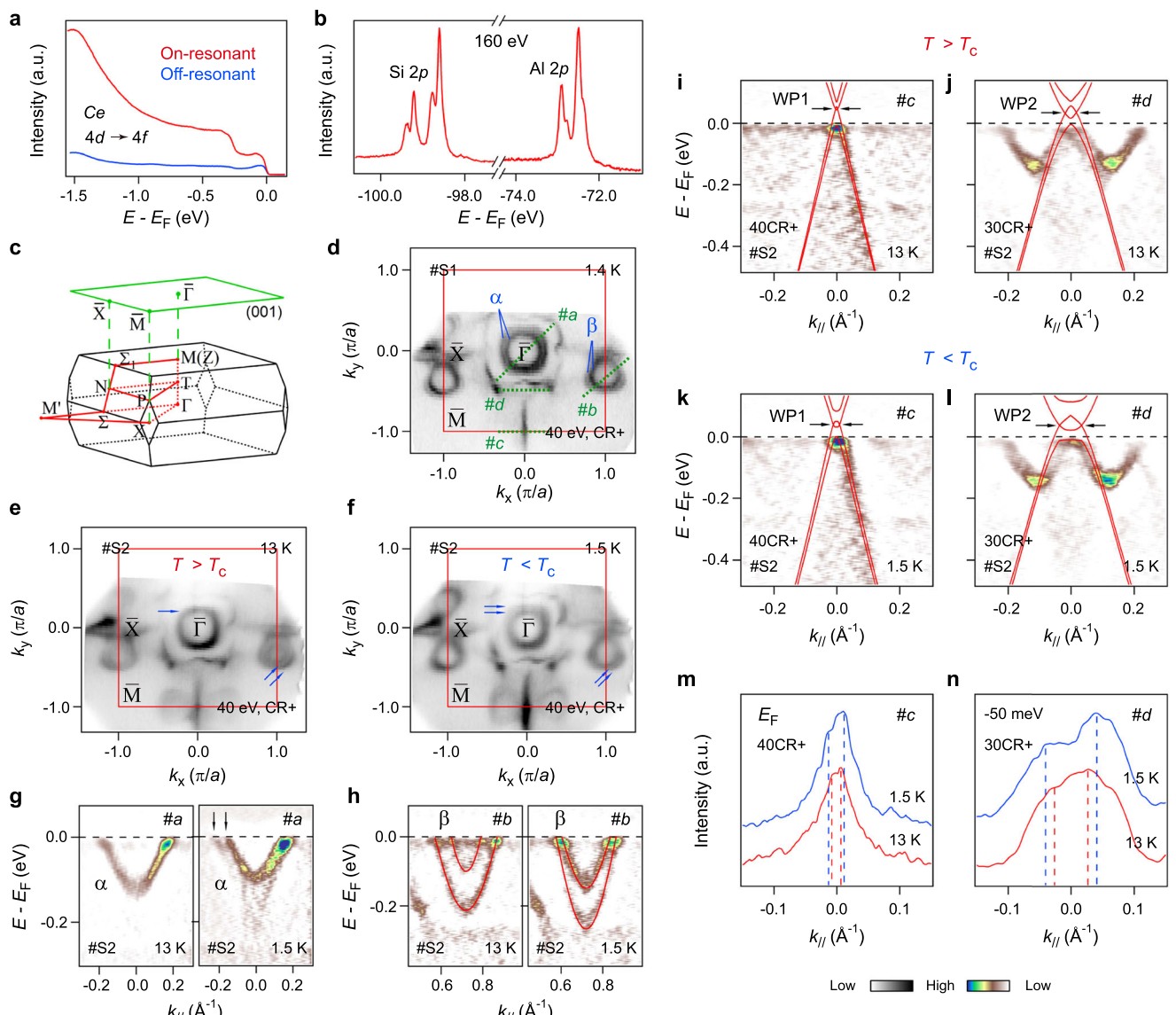

**Fig. 2 | ARPES measurements of CeAlSi. a** Angle-integrated photoemission spectra of CeAlSi with Ce $N$ edge on-resonant (121 eV) and off-resonant (116 eV) photons, respectively. **b** Core-level photoemission spectra of CeAlSi recorded at $h\nu = 160$ eV. **c** Sketches of 3D BZ and (001)-surface BZ for the noncentrosymmetric $I4_1/md$ space group structure. **d** Constant-energy ARPES image of CeAlSi ($h\nu = 40$ eV, CR$^+$ polarization, $T = 1.4$ K, sample no. S1) obtained by integrating the photoemission intensity within $E_F \pm 20$ meV. Cuts #a − #d indicate the locations of the band dispersions in **g**, **h**, **i**, **k**, **j**, **l**, respectively. **e**, **f** Constant-energy maps at $E_F$ of CeAlSi ($h\nu = 40$ eV, CR$^+$ polarization, sample no. S2) taken above (13 K) and below (1.5 K) $T_C$, respectively. The red solid curves in **d**–**f** represent the (001)-projected BZs. $a$ (= 4.26 Å) is the in-plane lattice constant of CeAlSi. **g** Second derivative

intensity plots of CeAlSi measured along cut #a above and below $T_C$, respectively. The black arrows indicate the splitting of $\alpha$ band below $T_C$. **h** Same as **g** recorded along cut #b. The red solid curves are guides to the eye for the split $\beta$ bands. **i**, **j** Second derivative intensity plots measured along cuts #c and #d above $T_C$, respectively. The appended red curves are the bulk band calculations. The black arrows indicate the WP1/WP2. **k**, **l** Same as **i**, **j** recorded below $T_C$. In **j** and **l**, the photon energy of 30 eV is utilized to better reveal the hole-like bands which host the WP2. **m** MDCs taken at $E_F$ of the raw data along cut #c. The dashed lines are guides to the eye for the peak positions. **n** Same as **m** taken at −50 meV of the raw data along cut #d.

extract the momentum distribution curves (MDCs) at $E_F$, as shown in Fig. 2m. One can see that the $\Delta k_F$ of the hole-like band, which hosts the WP1, increases upon entering the ferromagnetic state (from -0.012 to -0.027 Å$^{-1}$), agreeing well with the calculations. We also perform a similar analysis on the band structure along cut #d. In Fig. 2j, l (cut #d), the hole-like bands below $E_F$ match well with the calculations; the additional bands are the projections from other $k_z$ planes due to the $k_z$ broadening effect, as reported in PrAlSi[62]. Similarly, a pair of WP2 are located at about 35 meV above $E_F$ and move away from each other when entering the ferromagnetic state (from -0.035 to -0.071 Å$^{-1}$), which is further supported by the MDCs taken at 50 meV below $E_F$ [Fig. 2n].

## Pressure evolution of anomalous Hall effect and pressure-induced phase transitions in CeAlSi

Figure 3a displays the resistivity profiles at various pressures. Under pressure, $T_C$ increases monotonically with pressure up to 8.5 GPa [Fig. 3a], beyond which it cannot be resolved. To illuminate the pressure evolution of the ferromagnetic transition, we performed the $ac$ magnetic susceptibility measurements. As displayed in Fig. 3b, e, $T_C$ initially increases with pressure till ~10.3 GPa, and then decreases. The pressure evolution of $T_C$ deduced from $ac$ susceptibility is consistent with that from resistivity [Fig. 3e]. Through a careful comparison of them, we found pressure may tune the electron correlation effect as well as valence fluctuations (discussed later). Figure 3c shows the Hall

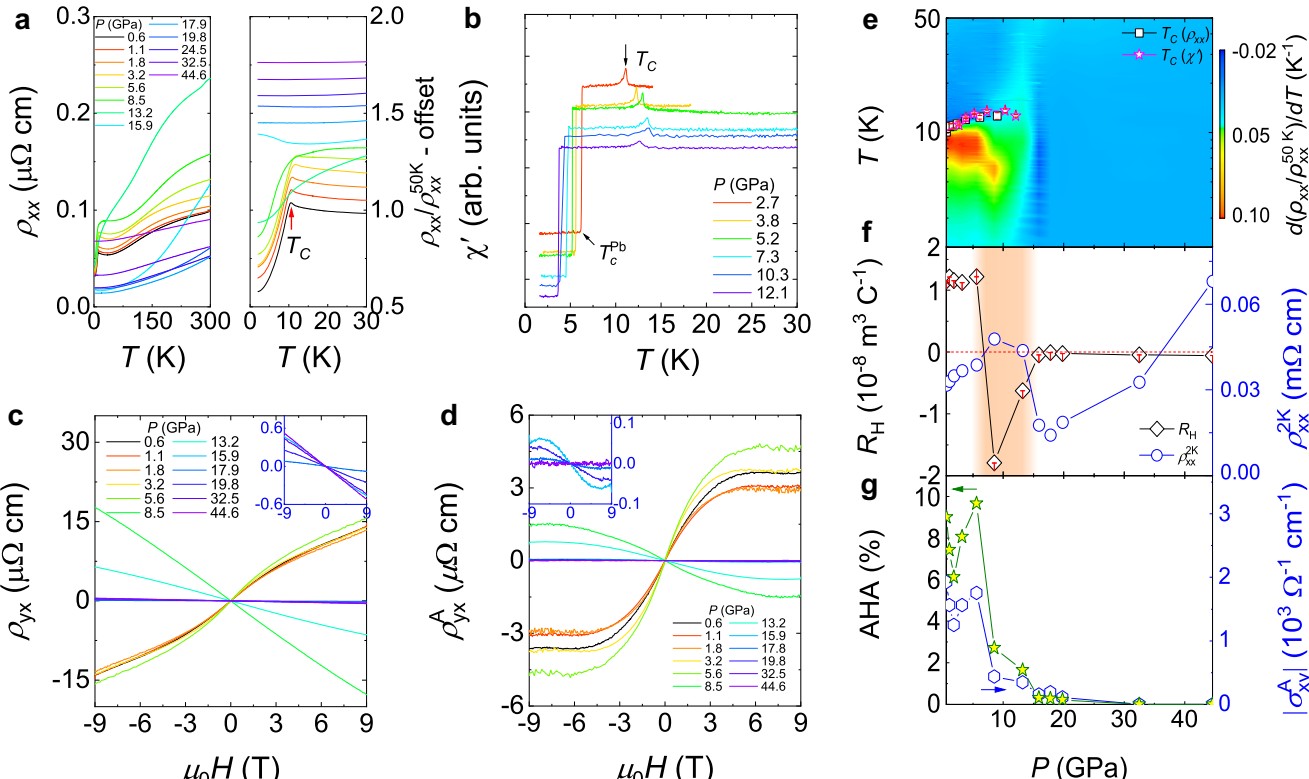

**Fig. 3 | Pressure-induced phase transitions in CeAlSi. a** Left profiles: temperature dependence of longitudinal resistivity at different pressures. Right profiles: low-temperature resistivity normalized to the data at 50 K. With increasing pressure, the ferromagnetic transition temperature ($T_C$) initially increases. **b** ac magnetic susceptibility under pressure. **c** Hall resistivity at various pressures. Above 5.6 GPa, the slope of Hall resistivity changes sign, indicating that the dominant carriers change from holes to electrons. Inset shows the high-pressure data above 15.9 GPa. **d** Anomalous Hall resistivity ($\rho_{yx}^A$) at various pressures. Inset shows the high-pressure data. **e** Contour plot of the derivative of normalized resistivity at different pressures. The background color represents the $d(\rho_{xx}/\rho_{xx}^{50K})/dT$ value. The pressure evolution of $T_C$ deduced from ac magnetic susceptibility and resistivity has been added. Error bars represent the full width at half max of peaks. **f** Pressure-dependent Hall coefficient ($R_H$) and the resistivity at 2 K ($\rho_{xx}^{2K}$). $R_H$ is obtained through linear fits to the high-field data. Error bars are deduced from the fit. The shaded area represents the pressure region where $R_H$ changes sign, suggesting the existence of a pressure-induced Lifshitz transition. **g** Pressure dependence of anomalous Hall angle (AHA) and an absolute value of anomalous Hall conductivity ($|\sigma_{xy}^A|$).

resistivity at 2 K at various pressures. With increasing pressure to 3.2 GPa, the Hall resistivity decreases slightly, followed by a slight enhancement at 5.6 GPa. Surprisingly, when the pressure reaches 8.5 GPa, the slope of the Hall resistivity changes sign abruptly, indicating a pressure-induced Lifshitz transition. The contour profiles of the derivative of the normalized resistivity with respect to temperature, the pressure evolutions of the Hall coefficient ($R_H$), and the resistivity at 2 K are plotted in Fig. 3e, f. As may be seen, the pressure-induced Lifshitz transition seems to correspond to the evolution of magnetism. We also calculated the magnetic moments under pressure via DFT calculations, yielding $0.8364\mu_B$, $0.9657\mu_B$, $0.6811\mu_B$, and $0.00256\mu_B$ for 0, 10, 20, and 40 GPa, respectively, which is overall consistent with the experimental data. Thus, the enhancement of $T_C$ under low pressure derives from the pressure-driven enhancement of magnetic moments. Under higher pressure, the magnetic moments decrease gradually, and then disappear, leading to a magnetic phase transition from the ferromagnetic to a paramagnetic state. Since we demonstrate that magnetism has a strong impact on the evolution of band structure at ambient pressure, thus the pressure evolution of magnetic moments provides a strong hint that the Lifshitz transition arises from the coupling between the electronic band structure and magnetic configurations. We obtain the anomalous Hall resistivity by subtracting the ordinary contribution, and the intrinsic AHE is also verified [Supplementary Fig. 5a]. Upon increasing pressure, $\rho_{yx}^A$ initially decreases, which is consistent with the previous study[45]. However, beyond 1.8 GPa, $\rho_{yx}^A$ increases, reaching a maximum of 5.6 GPa. At

8.5 GPa, a sign change from positive to negative accompanied by a slight reduction in $\rho_{yx}^A$ implies that the dominant carriers change from hole to electron. Upon further compression, $\rho_{yx}^A$ decreases monotonically and then cannot be resolved above 32.5 GPa. The pressure-dependent AHA and the absolute value of anomalous Hall conductivity $|\sigma_{xy}^A|$ are also calculated, as plotted in Fig. 3g, which have a similar evolution as $\rho_{yx}^A$ in Fig. 3d. The anomalous Hall angles are -9.0% and -9.7% for 0.6 and 5.6 GPa, respectively.

To obtain more information about the pressure-induced phase transitions, we investigate the pressure evolution of the crystal structures and electronic band structures. Figure 4a displays the high-pressure XRD profiles. Under pressure, the crystal structure with the space group of $I4_1md$ persists up to 39.3 GPa. Upon further compression, a new diffraction peak situated at ~10.9° arises, indicative of a structural phase transition. The emerged high-pressure phase coexists with the $I4_1md$ phase up ~60 GPa. Figure 4c shows the pressure evolution of lattice constants with respect to 1 GPa [Fig. 4b] extracted from Rietveld refinements. The ratio of $a/c$ is plotted in the lower panel of Fig. 4c. As can be seen, in addition to the structural phase transition, there are two anomalies at ~10 and ~20 GPa, which correspond with the pressures where the Lifshitz transition and the transition from the magnetic state to a paramagnetic state in resistivity appear, respectively. The band structures at several selected pressures are calculated, which remain overall unchanged (Supplementary Fig. 7), except that the hole pockets along the Γ−X line become smaller with pressure and then transform to electron

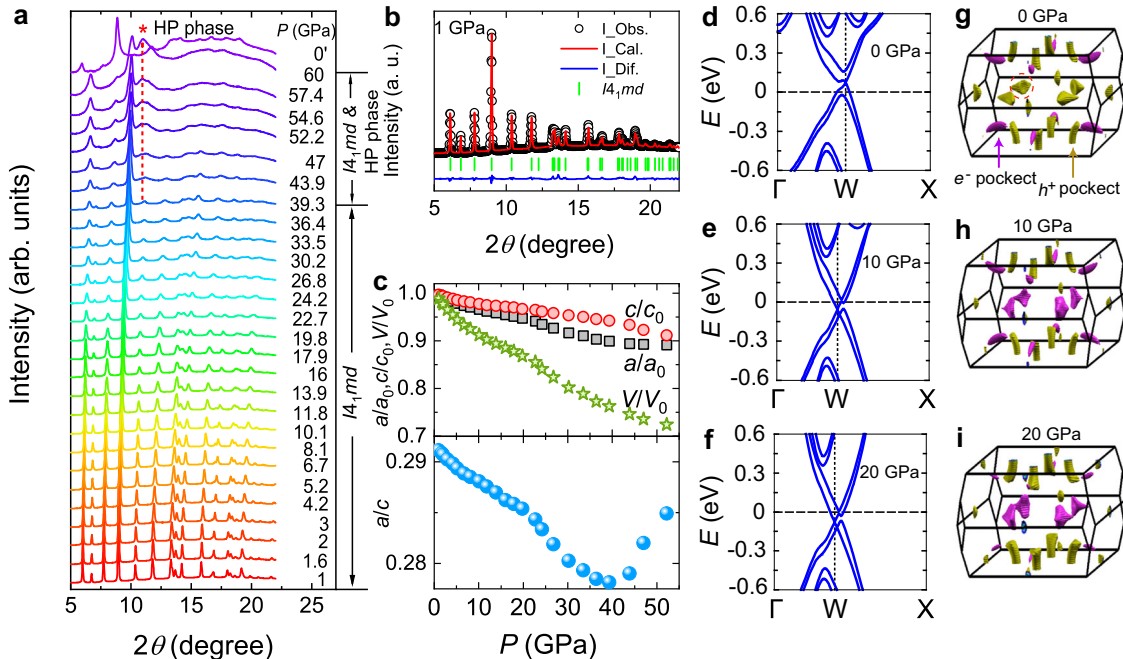

**Fig. 4 | Pressure evolution of the crystal structure and band structure of CeAlSi.**
**a** X-ray diffraction (XRD) pattern of CeAlSi at room temperature up to 60 GPa. The ambient-pressure structure with the space group of $I4_1md$ persists to -39.3 GPa, beyond which a new diffraction peak emerges (marked with a dashed line and asterisk), indicating that a pressure-induced structural phase transition occurs. 0′ represents that the pressure inside the sample chamber is released to zero, indicating that the emerging new structural phase is unstable at ambient pressure. **b** The Rietveld refinement of the XRD pattern at 1.0 GPa. The refined value is $R_P = 2.23\%$ with weighted profile $R_{WP} = 1.60\%$. The upper panel in (**c**) shows the

pressure-dependent normalized parameters $a/a_0$, $c/c_0$, and $V/V_0$ extracted from powder diffraction refinements. The lower panel in (**c**) shows the pressure evolution of the $a/c$ ratio. **d**–**f** Band structures of CeAlSi along the $\Gamma$–W–X line for 0, 10, and 20 GPa, respectively. **g**–**i** Calculated three-dimensional (3D) Fermi surfaces for 0, 10, and 20 GPa, respectively. The violet and dark yellow color represent electron pockets and hole pockets, respectively. At 10 GPa, pressure drives hole pockets (the red dashed circle as marked in **g**) into electron pockets, demonstrating the pressure-induced Lifshitz transition observed in Hall resistivity under pressure.

pockets at -10 GPa [Fig. 4d–i], which confirms the pressure-induced Lifshitz transition in CeAlSi. This also implies that the pockets along the $\Gamma$–X line dominate the transport behavior (the Hall coefficient under pressure changes from positive to negative) in CeAlSi. Since there is no distinct anomaly in the calculated band structures for 10 and 20 GPa, the structural anomalies probably arise from magnetostriction/magnetoelastic effects that are altered by pressure[45,47]. At 0 GPa, the Weyl nodes along the $\Gamma$–X line are located 74 meV above $E_F$, whereas they shift to −57 meV and −78 meV below $E_F$ for 10 and 20 GPa, respectively. This indicates that pressure tunes the crystal structure of CeAlSi, which consequently has an effect on the evolution of the Weyl nodes as well as, in turn, AHE. Based on our calculations, we found that pressure does not change the classification of Weyl nodes, but just shifts the positions of Weyl nodes as observed in the magnetism scenario (Supplementary Table 1).

## Discussion

Previous high-pressure studies on CeAlSi revealed a monotonic decline of AHE and LHE with pressure up to 2.7 GPa, while the negligible pressure effect on the magnetic structure and electronic band structure implies the importance of the nontrivial domain walls for the anomalous transport behavior of CeAlSi[43,45]. Under pressure, the anomalous Hall resistivity of CeAlSi is enhanced compared with that at ambient pressure. Considering that the dimensions of the sample (Supplementary Fig. 1) we used are comparable to the size of one single magnetic domain[41,44], the relative impact of the domain-wall landscapes or magnetostriction/magnetoelastic effects on topological properties in the high-pressure sample may be more pronounced than in the ambient-pressure scenario, since the two kinds of domain walls may have different interactions with Weyl fermions[43,45]. As a consequence, in addition to the tuning of the positions of Weyl nodes via

pressure, the contributions to AHE from the landscapes of domain walls need to be elaborated further. The local 4$f$-moments of $Ce^{3+}$ in CeAlSi interact within the lattice, leading to a noncollinear ferromagnetic ordering[41]. Previous ARPES experiments suggest that Ce 4$f$ electrons could play a role in CeAlSi, although the band deriving from Ce 4$f$ electrons is -0.3 eV below $E_F$[42]. Beyond the ferromagnetic transition, it's worth noting that other factors, such as Kondo coupling, the Ruderman–Kittel–Kasuya–Yosida (RKKY) interaction, the interplay between Kondo coupling and RKKY interaction, as well as valence fluctuations, may all be modulated by the application of pressure. For more comprehensive insights, please refer to Supplementary Note 9. Therefore, the amplified quantum fluctuations in CeAlSi may also be recognized as a new platform for the origin of novel topological states of matter and various quantum phase transitions[64].

In summary, we systematically studied the band structure and topological properties of the ferromagnetic Weyl semimetal CeAlSi through anomalous magneto-transverse transport, ARPES, and band calculations, demonstrating that the positions of Weyl nodes can be tuned via magnetism and pressure. At ambient pressure, the enhancement of AHE and ANE across $T_C$ stems from the increased distance of Weyl nodes with opposite chirality. This is evidenced by our ARPES experiments together with DFT calculations. The essential role of magnetism in tuning the bulk and surface band structure of CeAlSi is clearly revealed, distinguishing CeAlSi from other magnetic $R$AlPn siblings established thus far. Under pressure, multiple phase transitions are discovered. High-pressure band calculations reveal that pressure could also shift the positions of Weyl nodes, which is in line with the transverse transport measurements. Additionally, the electron correlation effect may also play a significant role in the transport behavior under pressure. These results suggest that ferromagnetic CeAlSi could serve as a fertile and tunable platform to explore novel

topological states, and the interplay among magnetism, topology, and electron correlations.

## Methods

For the growth of CeAlSi single crystals, a self-flux method was adopted, as described in the literature[32]. The as-grown single crystals were characterized by compositional analysis, Laue diffraction pattern, and x-ray diffraction (XRD) measurements [Supplementary Fig. 2]. The details of sample preparation for electrical and thermoelectrical transport measurements, high-pressure electrical transport and XRD measurements, ARPES measurements, and first-principles calculations can be found in the Supplementary Information.

## Data availability

The data that support the findings of this study are available from the corresponding authors upon request.

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

## Acknowledgements

This work is supported by the Deutsche Forschungsgemeinschaft (DFG, German Research Foundation) through the projects C03, C04, and C07 of the Collaborative Research Center SFB 1143 (project-ID 247310070), the Würzburg-Dresden Cluster of Excellence on Complexity and Topology in Quantum Matter—ct.qmat (EXC 2147, Project No. 390858490). E.C. acknowledges the financial support from the Alexander von Humboldt Foundation. W.Y. acknowledges the National Natural Science Foundation of China (Grant No. U1930401). S.L. acknowledges the National Natural Science Foundations of China (Grant No. 12174064), and the Ministry of Science and Technology of China (Grant No. 2022YFA1402203). W.Z. is supported by the Shenzhen Peacock Team Plan (KQTD20170809110344233) and the Bureau of Industry and Information Technology of Shenzhen through the Graphene Manufacturing Innovation Center (201901161514). Y.G. acknowledges research funds from the State Key Laboratory of Surface Physics and Department of Physics, Fudan University (Grant No. KF2019 06). Y.X. was supported by the National Natural Science Foundation of China (Grant No. 12204033). C.F. acknowledged DFG-Projektnummer (No. 392228380) and ERC Advanced Grant No. (742068) "TOP-MAT" for funding. J.-G.C. is supported by the National Natural Science Foundation of China (12025408). Part of the measurements were conducted at the Cubic Anvil Cell Station, the Synergetic Extreme Condition User Facility (SECUF).

## Author contributions

E.C. conceived the idea and designed the experiments. E.C., J.Y., W.X. and Y.G. prepared the single crystals. E.C. conducted magnetic susceptibility measurements and electrical transport measurements at ambient pressure, and thermoelectric measurements with the help of M.B. Y.W. helped with the data collection. E.C. and L.Y. were responsible for electrical transport experiments under pressure. L.Y. and W.Y. conducted the high-pressure XRD measurements and analysis. X.S. and W.Z. performed the DFT calculations for the pressure evolution of the electronic band structure. R.L. and A.F. conducted ARPES experiments. P.Y., B.W. and J.-G.C. conducted ac magnetic susceptibility measurements. U.B. performed the compositional analysis. Y.X. performed the DFT calculations for the pressure evolution of the magnetic moments. N.P. and D.C.P. helped with the orientation of the single crystals for thermoelectric measurements. E.C., L.Y., X.S. and R.L. analyzed the data. E.C., R.L., W.Y. and B.B. supervised the project. E.C., L.Y., X.S. and R.L. contributed equally to this work. E.C. wrote the paper with input from all coauthors.

## Funding

## Competing interests

The authors declare no competing interests.

## Additional information

[1]Leibniz Institute for Solid State and Materials Research (IFW-Dresden), 01069 Dresden, Germany. [2]Max Planck Institute for Chemical Physics of Solids, 01187 Dresden, Germany. [3]Center for High Pressure Science and Technology Advanced Research, 201203 Shanghai, China. [4]State Key Laboratory of Superhard Materials, Department of Physics, Jilin University, 130012 Changchun, China. [5]State Key Laboratory of Advanced Welding & Joining, Harbin Institute of Technology, 150001 Harbin, China. [6]Flexible Printed Electronics Technology Center, Harbin Institute of Technology (Shenzhen), 518055 Shenzhen, China. [7]Helmholtz-Zentrum Berlin für Materialien und Energie, Albert-Einstein-Straße 15, 12489 Berlin, Germany. [8]Joint Laboratory "Functional Quantum Materials" at BESSY II, 12489 Berlin, Germany. [9]School of Physical Science and Technology, ShanghaiTech University, 200031 Shanghai, China. [10]Beijing National Laboratory for Condensed Matter Physics and Institute of Physics, Chinese Academy of Sciences, 100190 Beijing, China. [11]School of Physical Sciences, University of Chinese Academy of Sciences, 100190 Beijing, China. [12]Institute for Applied Physics, University of Science and Technology Beijing, 100083 Beijing, China. [13]Key Laboratory of Polar Materials and Devices (MOE), School of Physics and Electronic Science, East China Normal University, 200241 Shanghai, China. [14]Institute of Solid State and Materials Physics, Technische Universität Dresden, 01069 Dresden, Germany. [15]State Key Laboratory of Surface Physics, and Department of Physics, Fudan University, 200438 Shanghai, China. [16]Collaborative Innovation Center of Advanced Microstructures, 210093 Nanjing, China. [17]Shanghai Research Center for Quantum Sciences, 201315 Shanghai, China. [18]Institute of Solid State and Materials Physics and Würzburg-Dresden Cluster of Excellence—ct.qmat, Technische Universität Dresden, 01062 Dresden, Germany. ✉e-mail: erjian_cheng@163.com; lourui09@gmail.com; yangwg@hpstar.ac.cn; B.Buechner@ifw-dresden.de

