## [Peer Review File · Nature Communications]

REVIEWER COMMENTS

Reviewer #1 (Remarks to the Author):

The manuscript “Tunable positions of Weyl nodes via magnetism and pressure in the ferromagnetic Weyl semimetal CeAlSi” by Erjian Cheng et al. is a study of spectroscopic and transport properties of CeAlSi. Namely, the authors measured the electrical and thermoelectrical coefficients, magnetization, as well as performed X-ray diffraction, angle resolved photoemission spectroscopy (ARPES) and band structure calculations. The experiments were conducted in a magnetic field and under hydrostatic pressure. The main conclusion drawn by the authors is that there is a difference in the position of the Weyl nodes between the paramagnetic and ferromagnetic phase of CeAlSi, and that the application of pressure induces some phase transitions (including Lifshitz one).

I appreciate the amount of work the authors have devoted to this study, but I think the conclusions presented are neither sufficiently novel nor well supported. The magnetic Weyl semimetals are in fact an interesting subject of studies and have been investigated quite intensively recently. For CeAlSi alone, there have been reported the ARPES results (in the paramagnetic phase) [A.P. Sakhya et al., Phys. Rev. Materials 7, L051202 (2023)], anomalous Hall effect (AHE) [H.-Y. Yang et al., Phys. Rev. B 103, 115143 (2021)] and anomalous Nernst effect (ANE) [M.S. Alam et al., Phys. Rev. B 107, 085102 (2023)]. Results of the resistivity and Hall measurements under hydrostatic pressure were also published [M.M. Piva et al., Phys. Rev. Research 5, 013068 (2023)]. Although the results presented here extend those previously reported, I do not think this warrants publication in Nature Communications. Moreover, I think that the main topic of the manuscript, i.e. a difference in the position or even number of Weyl nodes in different magnetic states of a Weyl semimetal, is something to be expected and has already been shown to happen [e.g. J. Gaudet et al., Nature Materials 20, 1650 (2021), H.-Y. Yang et al., Phys. Rev. B 103, 115143 (2021), M.S. Alam et al., Phys. Rev. B 107, 085102 (2023)]. Even if this was unexpected, I do not find the claimed shift of Weyl nodes positions well evidenced. The authors write that the anomalous transverse transport proves the shift of the positions of Weyl nodes in the ferromagnetic phase. However, the evolution of both the anomalous Hall and Nernst coefficients' evolution is rather smooth and both effects appear at $T \sim 50$ K, while Curie temperature $T_C \sim 8$ K. As the anomalous Hall and Nernst effects require time reversal symmetry breaking, their contribution depends on the magnetization [e.g. P. Puphal et al., Phys. Rev. Lett. 124, 017202 (2020)] and its variation can account for changes of the anomalous coefficients. On the other hand, the ARPES results are also told to indicate a shift of Weyl nodes, but its relation to a split of the alpha and beta bands is not well explained. Furthermore, another conclusion drawn from measurements under pressure, i.e. indication the electronic structure is affected by structural transitions and changes in the lattice constants, seems to be pretty obvious.

In addition, I have some other comments and questions:

- The authors say that $|\sigma^A_{xy}|$ vs. σ_{xx} is approximately constant, but data shown in supplementary Fig. 5 suggest that it is not the case. It seems that relative changes of σ_{xx} for both samples are comparable to the respective changes of σ^A_{xy} .

- The claim that pressure tunes the correlations between electron needs better support. Changes in the temperature dependence of the resistivity shown in supplementary Fig. 8, even if due to the Kondo effect, it do not necessarily imply that the electronic system is correlated.

- I believe that the authors generally tend to exaggerate their claims. For example, the last sentence of the abstract states that CeAlSi is catering to an array of potential applications, such as spintronics and thermoelectrics. While the former is in principle possible in some distant future, I do not really see a reason to use CeAlSi for cooling or power generation.

- Again, a claim that CeAlSi is most promising platform hosting magnetic tunability is rather too bold.

- I am not sure what the authors mean when they say that interplay between magnetism and topology remains mysterious.

- The authors write that the enhancement of anomalous transverse transport arises from a shift of the position of Weyl nodes rather than intermediate states. What intermediate states are they?

In conclusion, I think that presented results do not meet Nature Communications' standards in terms of novelty and provided claims are not sufficiently supported by the analysis. I therefore recommend rejection of the manuscript in question.

Reviewer #2 (Remarks to the Author):

The present paper reports electrical, thermoelectric, high-pressure transport, and ARPES investigation of CeAlSi. This is thorough investigation and implies the topological nature and transport behavior of CeAlSi. I feel the work is well conducted and summarized. I have a few suggestions which need to be addressed before further processing:

1. The universal scaling relation between the anomalous Hall conductivity and the longitudinal conductivity is provided in the paper. To get a more precise understanding of the intrinsic transport, ρ_{yx} vs. ρ_{xx}^2 fitting data and discussion should be added for both ambient and high-pressure conditions.
2. There is very little discussion about the anisotropic nature of the material. Can the authors provide more discussion on this point? The preliminary anisotropic electrical transport data (easy and hard axis) of the material should be added to the manuscript.
3. Please provide S_{xy} vs. magnetic field plot in the main text.
4. Please provide theoretically estimated anomalous Hall and Nernst conductivity as a function of Fermi energy.

5. Apart from PXR data, compositional analysis, and Laue diffraction data should be added.
6. Apart from topology, the presence of bipolarity i.e. two different types of charge carriers in a material can also cause a large Nernst effect. Can the authors comment on this considering the charge carrier types and transport nature in the present case?

Reviewer #3 (Remarks to the Author):

The manuscript systematically investigated the band structure and transport properties of the ferromagnetic Weyl semimetal CeAlSi at ambient and pressure conditions. At ambient pressure, an anomalous Hall effect (AHE) and an anomalous Nernst effect (ANE) arise in the paramagnetic state, and then are enhanced when the temperature approaches the ferromagnetic ordering temperature. Under a pressure, an enhancement and a sign change of AHE are discovered. These results indicate that magnetism and pressure can serve as efficient parameters to tune the positions of Weyl nodes in CeAlSi, evidenced by ARPES experiments and band calculation. Intuitively, the experiments are well done and the conclusion is also reasonable.

What my concerns are that the origins for the tunability of magnetism and pressure are still unresolved by these experiments.

1) Since the AHE and ANE arise from the paramagnetic state closing to the T_c , the authors attribute this to the magnetism tuning of the Weyl nodes. The ARPES measurements reveal that the position of beta-band with respect to E_F has also been adjusted via magnetism. Such an evolution of beta band dispersion indicates that magnetism tune the bulk band structure of CeAlSi as well as the positions of Weyl nodes. My question is that how the magnetism tunes this Weyl node in the paramagnetic state without FM long-range order?

2) The authors observed that there are band splittings of the alpha and beta FSs. The splitting of beta FS has been previously observed in PrAlSi and SmAlSi₆₀, while the split alpha FS has not been reported before in other RAIPn material. Because the beta band is related to the magnetism tuning of the Weyl nodes, which was also observed in other compounds, what is the new points or finding for CeAlSi? What is the role of the split alpha-FS? It sounds not to play a key role for the observed anomaly.

3) Figure 3(c) shows the Hall resistivity at 2 K at various pressures, the slope of the Hall resistivity changes sign abruptly at 5.6 GPa, indicating a pressure-induced Lifshitz transition. In Fig.3(f), the shaded area represents the pressure region where RH changes sign, but this pressure-induced LF transition seems to appear only at temperatures far below T_c as shown in Fig.3(e). The origins of magnetism and pressure tuning band structure sound very different. Which bands are sensitive to the pressure, the alpha and beta FSs?

4) In the discussion section, considering the dimensions of the samples used which are comparable to the size of one single magnetic domain, the authors said the magnetic texture as well as the topological properties can be affected by the domain-wall landscapes. To a single domain crystal, why the domain-wall landscapes can be a key factor? Especially, the AHE appears in the paramagnetic state above T_c

In summary, the experiments are well done, the data are high quality. But the origins for the observations are not convincing. I suggest the authors should make clear what are the new points compared to the other similar compounds, and clearly answer why this material presents different.

Reviewer #1 (Remarks to the Author):

The manuscript “Tunable positions of Weyl nodes via magnetism and pressure in the ferromagnetic Weyl semimetal CeAlSi” by Erjian Cheng *et al.* is a study of spectroscopic and transport properties of CeAlSi. Namely, the authors measured the electrical and thermoelectrical coefficients, magnetization, as well as performed X-ray diffraction, angle resolved photoemission spectroscopy (ARPES) and band structure calculations. The experiments were conducted in a magnetic field and under hydrostatic pressure. The main conclusion drawn by the authors is that there is a difference in the position of the Weyl nodes between the paramagnetic and ferromagnetic phase of CeAlSi, and that the application of pressure induces some phase transitions (including Lifshitz one).

I appreciate the amount of work the authors have devoted to this study, but I think the conclusions presented are neither sufficiently novel nor well supported. The magnetic Weyl semimetals are in fact an interesting subject of studies and have been investigated quite intensively recently. For CeAlSi alone, there have been reported the ARPES results (in the paramagnetic phase) [A.P. Sakhya *et al.*, *Phys. Rev. Materials* 7, L051202 (2023)], anomalous Hall effect (AHE) [H.-Y. Yang *et al.*, *Phys. Rev. B* 103, 115143 (2021)] and anomalous Nernst effect (ANE) [M.S. Alam *et al.*, *Phys. Rev. B* 107, 085102 (2023)]. Results of the resistivity and Hall measurements under hydrostatic pressure were also published [M.M. Piva *et al.*, *Phys. Rev. Research* 5, 013068 (2023)]. Although the results presented here extend those previously reported, I do not think this warrants publication in Nature Communications. Moreover, I think that the main topic of the manuscript, i.e. a difference in the position or even number of Weyl nodes in different magnetic states of a Weyl semimetal, is something to be expected and has already been shown to happen [e.g. J. Gaudet *et al.*, *Nature Materials* 20, 1650 (2021), H.-Y. Yang *et al.*, *Phys. Rev. B* 103, 115143 (2021), M.S. Alam *et al.*, *Phys. Rev. B* 107, 085102 (2023)]. Even if this was unexpected, I do not find the claimed shift of Weyl nodes positions well evidenced. The authors write that the anomalous transverse transport proves the shift of the positions of Weyl nodes in the ferromagnetic phase. However, the evolution of both the anomalous Hall and Nernst coefficients' evolution is rather smooth and both effects appear at $T \sim 50$ K, while Curie temperature $T_C \sim 8$ K. As the anomalous Hall and Nernst effects require time reversal symmetry breaking, their contribution depends on the magnetization [e.g. P. Puphal *et al.*, *Phys. Rev. Lett.* 124, 017202 (2020)] and its variation can account for changes of the anomalous coefficients. On the other hand, the ARPES results are also told to indicate a shift of Weyl nodes, but its relation to a split of the alpha and beta bands is not well explained. Furthermore, another conclusion drawn from measurements under pressure, i.e., indication the electronic structure is affected by structural transitions and changes in the lattice constants, seems to be pretty obvious.

Reply: We thank Reviewer #1 for these valuable comments. First and foremost, it is imperative to emphasize that this work represents a substantial advancement beyond prior research. In recent times, a significant amount of effort has been dedicated to exploring the intriguing realm of the $RAIPn$ family, unearthing a multitude of extraordinary phenomena. As pointed out by Reviewer #1, earlier investigations, employing techniques such as ARPES and magneto-transport measurements, have primarily centered on CeAlSi. In stark contrast, our focus transcends the characterization of the fundamental topological band structure (including Fermi arcs and Weyl nodes), magnetic properties (nonlinear magnetic structure and chiral magnetic domain wall), and magneto-transport (highlighting AHE and ANE). Our research delves into the intricate interplay between magnetism and topology, utilizing ARPES and magneto-transport methodologies. Distinguishing itself from the majority of magnetic Weyl semimetals, the breaking of time-reversal symmetry in $RAIPn$ simply serves as a Zeeman field, causing the shift of Weyl nodes' positions. This unique feature provides an exceptional platform for studying the effects of Weyl node movement on transport phenomena and even offers the potential for manipulating Weyl fermions by straightforward cooling of the system. Nevertheless, up to the present time, experimental verification of this remains outstanding, and such verification constitutes the central focus of our research. Additionally, we have also investigated the pressure effects in our study. In prior high-pressure investigations, M. M. Piva *et al.* demonstrated that the electronic band structure and magnetic properties are not markedly pressure-dependent, up to 2.7 GPa [M.M. Piva *et al.*, *Phys. Rev. Research* **5**, 013068 (2023)]. Similarly, W. Cao *et al.* also asserted the presence of a weak, negligible pressure effect on the ferromagnetic ground state and band structure [W. Cao *et al.*, *Chinese Phys. Lett.* **39**, 047501 (2022)]. These findings sharply contrast with our results. Employing high-pressure electrical transport, X-ray diffraction (XRD), and density functional theory (DFT) calculations, we have unveiled that pressure exerts a substantial influence on the tuning of topological properties, such as shifting the positions of Weyl nodes and affecting ferromagnetism. Moreover, it has the potential to induce phase transitions and potentially modulate electron correlation effects. Therefore, our work extends beyond mere reproduction, encompassing resistivity and Hall measurements under pressure.

The emergence of the anomalous Hall effect (AHE) and anomalous Nernst effect (ANE) in the paramagnetic state of CeAlSi does not necessitate spontaneous time-reversal symmetry breaking, as demonstrated by prior studies [J. Noky *et al.*, *Phys. Rev. B* **98**, 241106(R) (2018), T. Liang *et al.*, *Nat. Phys.* **14**, 451 (2018)]. These effects can be attributed to the existence of a nonzero Berry curvature, which is generated by Weyl nodes resulting from the breaking of space-inversion (SI) symmetry. It is worth noting that our work and the study conducted by M.S. Alam *et al.*, which reported the anomalous Nernst effect (ANE) [*Phys. Rev. B* **107**, 085102 (2023) as mentioned by Reviewer #1], were conducted simultaneously. This concurrent aspect has been duly acknowledged at the conclusion of our manuscript, and our research has also been cited by M.S. Alam *et al.* (Ref. 55).

Figure R1. **a** Constant-energy ARPES image of CeAlSi obtained by integrating the photoemission intensity within $E_F \pm 20$ meV (adopted from Fig. 2d in the main text). Cuts #c and #d indicate the locations of the band dispersions in **b,c** and **d,e**, respectively. **b,c** Second derivative intensity plots of CeAlSi measured along cut #c above and below T_c , respectively. The appended red curves are the bulk band calculations. The black arrows indicate the WP1. **d,e** Same as **b,c** recorded along cut #d. The black arrows indicate the WP2. **f** MDCs taken at E_F of the raw data along cut #c. The dashed lines are guides to the eye for the peak positions. **g** Same as **f** taken at -50 meV of the raw data along cut #d.

To delve further into the tunability of Weyl nodes, as raised by Reviewer #1, we conducted additional ARPES measurements together with DFT analysis. In the realm of band structure analysis, previous investigations into the evolution of the band structure throughout the ferromagnetic phase transition in CeAlSi have predominantly remained theoretical. The sole ARPES study available [*Phys. Rev. Materials* 7, L051202 (2023)] is confined to the paramagnetic phase. Therefore, to the best of our knowledge, our ARPES study represents the inaugural experimental assessment of the band structures of CeAlSi both above and below its T_c . What's particularly noteworthy is that while theoretical calculations have foreseen the potential for magnetism-induced alterations in the electronic structures of $RAIPn$, including the relocation of Weyl nodes, the actual manifestations in real cases prove to be considerably intricate. This is underscored by the temperature-independent electronic structures observed in PrAlSi and SmAlSi, as referenced in Ref. 61. Prior to our current research, there had been a dearth of direct experimental demonstrations in this regard, rendering CeAlSi an exceedingly rare exemplar within the $RAIPn$ family.

In the previous version, we detailed the alteration of both the α and β bands due to the presence of

ferromagnetism, providing strong evidence that CeAlSi indeed exhibits magnetism-tunable electronic structures. Consequently, when compared to other $RAIPn$ compounds, CeAlSi stands out as a highly promising platform for the control of Weyl node positions through magnetic manipulation. However, in response to Reviewer #1's concerns, we recognize that the relationship between the shift of Weyl nodes and the evolution of the α and β bands is not straightforward. In order to more comprehensively bolster the assertion of magnetic tunability of Weyl nodes, we have now conducted an in-depth examination of the band dispersions constituting the Weyl points, both above and below T_C .

As illustrated in Fig. R1(a), we examined the temperature evolution of two pairs of Weyl points (denoted as WP1 and WP2) along cuts #c and #d, respectively. In Figs. R1(b,c) (cut #c), one obtains a good consistency between experiments and calculations below E_F . As guided by the calculations, a pair of WP1 are located at about 50 ($T > T_C$) and 40 ($T < T_C$) meV above E_F ; it is notable that the distance of WP1 with opposite chirality greatly increases as entering the ferromagnetic phase (from ~ 0.006 to $\sim 0.020 \text{ \AA}^{-1}$). To validate that the WP1 moves away from each other due to the ferromagnetism, we extracted the momentum distribution curves (MDCs) at E_F , as shown in Fig. R1(f). One can see that the Δk_F of the hole-like band, which hosts the WP1, increases upon entering the ferromagnetic state (from ~ 0.012 to $\sim 0.027 \text{ \AA}^{-1}$), agreeing well with the calculations. We also performed similar analysis on the band structure along cut #d, as shown in Figs. R1(d,e,g). In Figs. R1(d,e), the hole-like bands below E_F matches well with the calculations; the additional bands are the projections from other k_z planes due to the k_z broadening effect, as reported in PrAlSi (Ref. 61). Similarly, a pair of WP2 are located at about 35 meV above E_F and move away from each other when entering the ferromagnetic state [from ~ 0.035 to $\sim 0.071 \text{ \AA}^{-1}$, Figs. R1(d-e)], which is further supported by the MDCs taken at 50 meV below E_F (Fig. R1g).

We would like to point out that, it would be nicer if we can tune the Weyl points to E_F or below E_F in ARPES measurements by in-situ dispensing of alkali-metal atoms to the sample surface to introduce electron doping, however, we find that similar to the case of PrAlSi and SmAlSi we studied before, the in-situ cleaved CeAlSi samples show quite rapid degradation (aged within 2 hours under a vacuum of $\sim 6 \times 10^{-11}$ Torr). As a result, with such sensitive sample surface, it is almost unfeasible to get convincing results after the in-situ alkali metal dosing. Therefore, based on these new results and analysis, we can conclude that CeAlSi hosts the magnetically tunable Weyl nodes near E_F , which have hitherto not been reported in other magnetic $RAIPn$ siblings, rendering CeAlSi itself unique. To clarify this point, we have added these data and discussions in the revised manuscript.

Regarding our measurements under pressure, we concur with the observations made by Reviewer #1, recognizing the evident interplay between crystal structure evolution and its consequential impact on electronic structure. Nevertheless, it's worth emphasizing that the influence of pressure can vary significantly across different systems. Within the realm of topological materials, pressure serves as a

highly valuable tool for fine-tuning both crystal and band structures, giving rise to a spectrum of intriguing physical phenomena, such as superconductivity, metal-insulator transitions, and topological phase transitions, among others. In the specific case of CeAlSi, prior high-pressure experiments have indeed been conducted. However, the precise mechanisms and the extent to which pressure affects the evolution of the topological band structure have remained somewhat unclear. This knowledge gap has been a primary driver motivating our high-pressure measurements in this study.

Furthermore, it's noteworthy that Ce-based compounds are often associated with electron correlation effects, a topic extensively explored in the context of Ce-based heavy fermions [Z. F. Weng *et al. Rep. Prog. Phys.* **79**, 094503(2016)]. Typically, due to their low energy scales, the ground states of Ce-based heavy fermion systems can be readily manipulated by the application of pressure, leading to the emergence of novel phenomena, including superconductivity and quantum critical points. This intriguing characteristic has also been a driving force behind our investigation into the evolution of CeAlSi's ground state under pressure. Surprisingly, in contrast to the majority of Ce-based materials, where magnetic transitions tend to diminish under pressure [Z. F. Weng *et al. Rep. Prog. Phys.* **79**, 094503(2016)], we observed that ferromagnetism in CeAlSi becomes notably enhanced when subjected to pressure, ultimately ceasing abruptly. This abrupt transition leads to a magnetic transformation from a ferromagnetic to a paramagnetic state. The shift from a magnetically ordered state to a normal metallic state in CeAlSi hints at possible valence fluctuations [D. I. Khomskii, *Transition metal compounds*. 501 (2014)], rendering CeAlSi an extraordinary platform for the investigation of topological properties, the intricate interplay between magnetism and topology, as well as the effects of electron correlation.

In addition, I have some other comments and questions:

1) The authors say that $|\sigma^{A}_{xy}|$ vs. σ_{xx} is approximately constant, but data shown in supplementary Fig. 5 suggest that it is not the case. It seems that relative changes of σ_{xx} for both samples are comparable to the respective changes of σ^{A}_{xy} .

Reply: We thank Reviewer #1 for pointing this out. Regarding S1, we plotted $|\sigma^{A}_{xy}|$ vs. σ_{xx} at different temperatures, in a manner consistent with the profiles of other reference materials, and the data demonstrates an approximate constancy, akin to materials such as MnSi, etc. However, for S2, we have exclusively presented data obtained at 2 K under different pressure conditions. Given that the crystal structure as well as the band structure vary at different pressures, $|\sigma^{A}_{xy}|$ displays a relative change in relation to σ_{xx} . Nonetheless, it's important to note that the $|\sigma^{A}_{xy}|$ vs. σ_{xx} profiles for both samples ultimately fall within the $|\sigma^{A}_{xy}| \sim \text{const.}$ regime. To ensure utmost clarity, we have updated the caption of Supplementary Fig. 5 accordingly.

2) The claim that pressure tunes the correlations between electron needs better support. Changes in the temperature dependence of the resistivity shown in supplementary Fig. 8, even if due to the Kondo effect, it do not necessarily imply that the electronic system is correlated.

Reply: We extend our gratitude to Reviewer #1 for this insightful comment. When subject to pressure, the ground state of CeAlSi undergoes significant changes, resulting in non-monotonic behavior in the resistivity, as illustrated in Fig. 3. In addition to the ferromagnetic transition, it's important to consider the potential influence of other factors, such as the Kondo effect and valence fluctuations. However, the application of pressure has the effect of broadening the profiles of resistivity, which complicates the analysis of the data. To gain further insight into this complex behavior, we conducted ac susceptibility measurements under high pressure. As depicted in Fig. R2(a), the magnetic transition initially strengthens under the influence of pressure, only to gradually diminish thereafter. The critical temperature, T_C , reaches a peak value of approximately 10 GPa, as indicated in the inset to Fig. R2(a). In Fig. R2(b), we present the low-temperature resistivity data, and the T_C values obtained from ac susceptibility measurements [determined through linear interpolation of the data in the inset of Fig. R2(a)] are marked for the sake of comparison. Notably, beyond the 5.6 GPa mark, the low-temperature resistivity exhibits distinct behavior, as shown in Fig. R2(b). At 8.5 GPa, in addition to the magnetic transition, an additional broad peak emerges (highlighted with a black arrow). This profile bears a resemblance to observations in certain Ce-based heavy fermion systems characterized by valence fluctuations [Z. F. Weng *et al. Rep. Prog. Phys.* **79**, 094503(2016); Z. Ren *et al., Phys. Rev. X* **4**, 031055(2014); G. W. Scheerer *et al., npj Quantum Mater.* **3**, 41 (2018)].

To further address the electron correlation effect, we fitted the low-temperature data at different pressures, using $\rho = \rho_0 + AT^n$, as shown in Fig. R2(c). Here, the ρ_0 , A , and n represent the residual resistivity, the inelastic electron-electron scattering coefficient, and the exponentiation of temperature, respectively. For a Fermi-liquid system, n is 2. The inelastic electron-electron scattering is enormously stronger (by three or four orders of magnitude) in heavy-fermion materials than it is in normal metals [R. Joynt *et al., Rev. Mod. Phys.* **74**, 235 (2002)]. Typically, it is governed by the density of states at the Fermi energy and a simple relation $\gamma_N \sim \sqrt{A}$ is fairly well obeyed across the family of compounds [R. Joynt *et al., Rev. Mod. Phys.* **74**, 235 (2002)]. From the fit, one can see that the n we obtained is slightly larger than 2, but far lower than 5 that from electron-phonon scattering in the low-temperature limit. Thus, CeAlSi is believed to be in the Fermi-liquid regime at all pressures. However, strikingly, the inelastic electron-electron scattering coefficient A displays a nonmonotonic evolution with pressure. It initially increases, and then peaks at ~ 8.5 GPa with a value of $0.216 \pm 0.006 \mu\Omega \text{ cm K}^{-1}$. This value is

comparable with those in heavy-fermion systems, for example, $0.55 \pm 0.05 \mu\Omega \text{ cm K}^{-2}$ in the typical heavy fermion superconductor UPt_3 for a current parallel to the hexagonal c axis [R. Joynt *et al.*, *Rev. Mod. Phys.* **74**, 235 (2002)]. Under higher pressure, the A decreases down to $0.00028 \pm 0.00002 \mu\Omega \text{ cm K}^{-n}$ (24.5 GPa), three orders of magnitude smaller than that at 8.5 GPa, which means that higher pressures drive CeAlSi into a normal metal. The pressure evolution of A agrees well with the magnetic transition under pressure [Fig. R2(a)], and it also reflects that pressure not only tune the magnetic transition but also the electron-electron correlation effect. In addition, according to the Doniach phase diagram, the Kondo interaction and the Ruderman-Kittel-Kasuya-Yoshida (RKKY) interaction compete with each other [S. Doniach, *Physica B+ C* **91**, 231-234 (1977)], and the balance between them would lead to a quantum critical point. However, this is not observed in CeAlSi. Based on the periodic Anderson model, the valence fluctuations can no longer be neglected [Z. F. Weng *et al. Rep. Prog. Phys.* **79**, 094503(2016)]. With continues tuning (herein, pressure), the system may be driven into a mixed valence regime, and a normal metal regime thereafter [D. I. Khomskii, *Transition metal compounds*. 501 (2014)], which is in line with what we observed in CeAlSi.

Figure R2. **a** ac susceptibility measurements under pressure. **b** Low-temperature resistivity normalized to the data at 50 K. Red arrows represent the evolution of ferromagnetic transition with pressure deduced from ac susceptibility. The black arrow for 8.5 GPa indicates another peak in resistivity. **c** The yielded parameters by fitting the low-temperature resistivity using $\rho = \rho_0 + AT^n$.

3) I believe that the authors generally tend to exaggerate their claims. For example, the last sentence of the abstract states that CeAlSi is catering to an array of potential applications, such as spintronics and thermoelectrics. While the former is in principle possible in some distant future, I do not really see a reason to use CeAlSi for cooling or power generation.

Reply: We would like to express our gratitude to Reviewer #1 for this valuable comment. The transverse thermoelectric properties of topological semimetals hold great promise for thermoelectric applications, as indicated in previous studies [C. G. Fu *et al.*, *APL Mater.* **8**, 040913 (2020); X. H. Chen, *Sci. China-*

Phys. Mech. Astron. **63**, 237031 (2020)]. In this study, we have successfully demonstrated that the manipulation of Weyl node positions in CeAlSi results in the enhancement of the anomalous Nernst signal. We propose that this discovery could potentially serve as a pivotal mechanism for the development of materials tailored for future thermoelectric applications.

4) Again, a claim that CeAlSi is most promising platform hosting magnetic tunability is rather too bold. - I am not sure what the authors mean when they say that interplay between magnetism and topology remains mysterious.

Reply: As previously discussed, it's crucial to acknowledge that CeAlSi stands as a unique case within the *RAIPn* compounds, as it is one of the rare instances where the experimentally verified tunability of Weyl nodes through magnetism has been established. In our revised manuscript, we have refined the related claim to emphasize that " compared to other *RAIPn* compounds discovered so far, CeAlSi is a very promising platform that hosting the magnetic tunability..." enhancing the rigor of our statement.

5) The authors write that the enhancement of anomalous transverse transport arises from a shift of the position of Weyl nodes rather than intermediate states. What intermediate states are they?

Reply: We sincerely appreciate Reviewer #1 for his/her valuable comment. In the introduction section, we initially discussed the magnetic field's potential to interact with magnetic moments, causing these moments to flip and align with the external field. Such interactions have been observed in various materials, including $\text{Mn}(\text{Bi}_{1-x}\text{Sb}_x)_2\text{Te}_4$, EuCd_2Sb_2 , EuAs_3 , etc., leading to the emergence of diverse topological states. Notably, in these cases, the possible enhancement of the anomalous Hall effect (if it exists) may arise from Berry curvature within the field-induced Weyl state. It is crucial to highlight that most of these materials are antiferromagnetic. In contrast, CeAlSi is a ferromagnetic material, and it has been proposed that the nonlinear spin configurations within it are robust against external magnetic fields up to 8 T. Therefore, the observed enhancement of the anomalous Hall effect (AHE) and anomalous Nernst effect (ANE) in CeAlSi can be attributed solely to the shift in the positions of Weyl nodes and not to other field-induced topological states, such as the field-driven topological nodal-line metal state, as observed in materials like EuAs_3 .

6) In conclusion, I think that presented results do not meet Nature Communications' standards in terms of novelty and provided claims are not sufficiently supported by the analysis. I therefore recommend rejection of the manuscript in question.

Reply: We would like to reiterate our appreciation to Reviewer #1 and are confident that we have

diligently addressed all the comments and concerns.

Reviewer #2 (Remarks to the Author):

The present paper reports electrical, thermoelectric, high-pressure transport, and ARPES investigation of CeAlSi. This is thorough investigation and implies the topological nature and transport behavior of CeAlSi. I feel the work is well conducted and summarized. I have a few suggestions which need to be addressed before further processing:

Reply: We thank Reviewer #2 for his/her positive evaluation on our work.

1) The universal scaling relation between the anomalous Hall conductivity and the longitudinal conductivity is provided in the paper. To get a more precise understanding of the intrinsic transport, ρ_{yx}^A vs. ρ_{xx}^2 fitting data and discussion should be added for both ambient and high-pressure conditions.

Figure R3. ρ_{yx}^A as a function of ρ_{xx}^2 for S1 at ambient pressure, and the fitting by using the Tian-Ye-Jin (TYJ) model, $\rho_{yx}^A = a\rho_{xx0} + b\rho_{xx}^2$, where the first and the second terms on the right-hand side of the equation represent the extrinsic and intrinsic contributions, respectively.

Reply: We appreciate Reviewer #2 for this insightful comment. We plot the ρ_{yx}^A vs. ρ_{xx}^2 for S1 at ambient pressure. As displayed in Fig. R3, the data in the ferromagnetic state there are two regimes that can be well fitted using the TYJ model [Shen *et al.*, *APL Mater.* **10**, 090705(2022)], i.e., one is in the ferromagnetic state, and the other one is across the ferromagnetic transition up to 12 K. The linear relationship between ρ_{yx}^A and ρ_{xx}^2 indicates the intrinsic contribution of AHE. Above 15 K,

ρ_{yx}^A vs. ρ_{xx}^2 does not follow a linear dependence, which means that AHE maybe comes from extrinsic contributions. Under pressure, it is not feasible to generate ρ_{yx}^A vs. ρ_{xx}^2 profiles, as we have exclusively measured the Hall resistivity at 2 K for all pressure conditions. Furthermore, combining 2 K data at different pressures for both band structures and magnetic configurations is unwarranted, given that pressure-induced modifications have been observed in both aspects. Given that we have effectively demonstrated the intrinsic contribution of the anomalous Hall effect (AHE) in CeAlSi (as depicted in Fig. R3), we have opted to employ the universal scaling relation between anomalous Hall conductivity and longitudinal conductivity, as presented in Supplementary Fig. 5(a). This allows us to elucidate the intrinsic AHE under varying pressures.

We thank Reviewer #2 again, and the Supplementary note 5 has been updated accordingly, and additional discussions have been incorporated as well.

2) There is very little discussion about the anisotropic nature of the material. Can the authors provide more discussion on this point? The preliminary anisotropic electrical transport data (easy and hard axis) of the material should be added to the manuscript.

Reply: We thank Reviewer #2 for this valuable suggestion. In the updated manuscript, the electrical and thermoelectrical transport results with magnetic field applied in plane are added (Supplementary Note 4).

3) Please provide S_{xy} vs. magnetic field plot in the main text.

Reply: The S_{xy} vs. magnetic field plot is plotted in Fig. 1 in the main text. Anomalous Hall resistivity and the Seebeck signal profiles are also moved to Fig. 1 from Supplementary Note 3.

4) Please provide theoretically estimated anomalous Hall and Nernst conductivity as a function of Fermi energy.

Reply: We appreciate Reviewer #2 for this valuable comment. The theoretically estimated anomalous Hall and Nernst conductivity as a function of Fermi energy have been added to the updated Supplementary Fig. 3.

5) Apart from P-XRD data, compositional analysis, and Laue diffraction data should be added.

Reply: We express our gratitude to Reviewer #2 for the valuable suggestion. The compositional analysis and the Laue diffraction data collected on the (001) plane have been added in the updated Supplementary

Fig. 2.

6) Apart from topology, the presence of bipolarity i.e. two different types of charge carriers in a material can also cause a large Nernst effect. Can the authors comment on this considering the charge carrier types and transport nature in the present case?

Reply: We are grateful to Reviewer #2 for this valuable comment. In the multiband metal NbSe₂, Nernst coefficient peaks at the temperature where the Hall coefficient switches sign, which was attributed to the ambipolar transport of compensated electron and hole bands [R. Bel *et al.*, *Phys. Rev. Lett.* **91**, 066602(2003)]. A similar phenomenon has also been observed in the Kagome lattice material CsV₃Sb₅ [Y. Gan *et al.*, *Phys. Rev. B* **104**, L180508 (2021)]. However, in the case of CeAlSi, apart from the anomalous contribution, the Hall resistivity can be accurately described by a single-band model, and no indication of a change in the sign of the Hall coefficient around T_C has been identified.

Reviewer #3 (Remarks to the Author):

The manuscript systematically investigated the band structure and transport properties of the ferromagnetic Weyl semimetal CeAlSi at ambient and pressure conditions. At ambient pressure, an anomalous Hall effect (AHE) and an anomalous Nernst effect (ANE) arise in the paramagnetic state, and then are enhanced when the temperature approaches the ferromagnetic ordering temperature. Under a pressure, an enhancement and a sign change of AHE are discovered. These results indicate that magnetism and pressure can serve as efficient parameters to tune the positions of Weyl nodes in CeAlSi, evidenced by ARPES experiments and band calculation. Intuitively, the experiments are well done and the conclusion is also reasonable.

What my concerns are that the origins for the tunability of magnetism and pressure are still unresolved by these experiments.

1) Since the AHE and ANE arise from the paramagnetic state closing to the T_C , the authors attribute this to the magnetism tuning of the Weyl nodes. The ARPES measurements reveal that the position of beta-band with respect to E_F has also been adjusted via magnetism. Such an evolution of beta band dispersion indicates that magnetism tune the bulk band structure of CeAlSi as well as the positions of Weyl nodes. My question is that how the magnetism tunes this Weyl node in the paramagnetic state without FM long-range order?

Reply: We sincerely appreciate Reviewer #3 for this valuable comment. In the case of CeAlSi, Weyl states are present in the paramagnetic phase due to the inversion symmetry breaking, and the anomalous

Hall and Nernst effects stem from the Berry curvature, as discussed in previous studies [C. G. Fu *et al.*, APL Mater. 8, 040913 (2020); X. H. Chen, Sci. China-Phys. Mech. Astron. 63, 237031 (2020)]. As the temperature approaches the ferromagnetic transition, both the anomalous Hall effect and the anomalous Nernst effect are intensified, primarily attributed to the magnetism-driven shift in the positions of the Weyl nodes. In the updated manuscript [Figs. 2(i-n)], we provided more compelling evidence (also see Fig. R1) to verify this.

2) The authors observed that there are band splittings of the alpha and beta FSs. The splitting of beta FS has been previously observed in PrAlSi and SmAlSi, while the split alpha FS has not been reported before in other $RAIPn$ material. Because the beta band is related to the magnetism tuning of the Weyl nodes, which was also observed in other compounds, what is the new points or finding for CeAlSi? What is the role of the split alpha-FS? It sounds not to play a key role for the observed anomaly.

Reply: We appreciate Reviewer #3 for this valuable comment. The difference between CeAlSi and other $RAIPn$ compounds is that the experimental band structure near E_F in the former is clearly modified by the emergence of ferromagnetism, while the band structures of the other compounds discovered so far have no noticeable evolution across the magnetic transitions. Most evidently, as the ferromagnetism develops in CeAlSi, the α band splits into two and the splitting scale of the β band changes.

Like the case in $Co_3Sn_2S_2$ [Q. Wang *et al.*, Nat. Commun. 9, 3681 (2018)], here the observed enhancement of AHE/ANE in the ferromagnetic state can be attributed to the increase of the distance of Weyl points with opposite chirality. To further support this argument, we have now studied the band dispersions composing the Weyl points above and below T_c . As illustrated in Fig. R1(a), we examined the temperature evolutions of two pairs of Weyl points (denoted as WP1 and WP2) along cuts #c and #d, respectively. In Figs. R1(b,c) (cut #c), one obtains a good consistency between experiments and calculations below E_F . As guided by the calculations, a pair of WP1 are located at about 50 ($T > T_c$) and 40 ($T < T_c$) meV above E_F ; it is notable that the distance of WP1 with opposite chirality greatly increases as entering the ferromagnetic phase (from ~ 0.006 to $\sim 0.020 \text{ \AA}^{-1}$). To validate that the WP1 moves away from each other due to the ferromagnetism, we extracted the momentum distribution curves (MDCs) at E_F , as shown in Fig. R1(f). One can see that the Δk_F of the hole-like band, which hosts the WP1, increases upon entering the ferromagnetic state (from ~ 0.012 to $\sim 0.027 \text{ \AA}^{-1}$), agreeing well with the calculations. We also performed similar analysis on the band structure along cut #d, as shown in Figs. R1(d,e,g). In Figs. R1(d,e), the hole-like bands below E_F matches well with the calculations; the additional bands are the projections from other k_z planes due to the k_z broadening effect, as reported in PrAlSi (Ref. 61). Similarly, a pair of WP2 are located at about 35 meV above E_F and move away from each other when entering the ferromagnetic state [from ~ 0.035 to $\sim 0.071 \text{ \AA}^{-1}$, Figs. R1(d,e)], which is further supported

by the MDCs taken at 50 meV below E_F [Fig. R1(g)].

We would like to point out that, it would be nicer if we can tune the Weyl points to E_F or below E_F in ARPES measurements by in-situ dispensing of alkali-metal atoms to the sample surface to introduce electron doping, however, we find that similar to the case of PrAlSi and SmAlSi we studied before, the in-situ cleaved CeAlSi samples show quite rapid degradation (aged within 2 hours under a vacuum of $\sim 6 \times 10^{-11}$ Torr). As a result, with such sensitive sample surface, it is almost unfeasible to get convincing results after the in-situ alkali metal dosing.

Therefore, based on these new results and analysis, we can conclude that CeAlSi hosts the magnetically tunable Weyl nodes near E_F , which have hitherto not been reported in other magnetic $RAlPn$ siblings, rendering CeAlSi itself unique. To clarify this point, we have added these data and discussions in the revised manuscript.

3) Figure 3(c) shows the Hall resistivity at 2 K at various pressures, the slope of the Hall resistivity changes sign abruptly at 5.6 GPa, indicating a pressure-induced Lifshitz transition. In Fig.3(f), the shaded area represents the pressure region where R_H changes sign, but this pressure-induced LF transition seems to appear only at temperatures far below T_C as shown in Fig.3(e). The origins of magnetism and pressure tuning band structure sound very different. Which bands are sensitive to the pressure, the α and β FSs?

Reply: The α band is of surface state origin and the β band probably contains both the bulk and surface components. Based on our current results, the pressure noticeably tunes the crystal structure of CeAlSi. As a result, the overall band structure (both bulk and surface) could be modified by the pressure, including the bands composing the Weyl nodes along the Γ -X direction [Figs. 4(d-f)].

4) In the discussion section, considering the dimensions of the samples used which are comparable to the size of one single magnetic domain, the authors said the magnetic texture as well as the topological properties can be affected by the domain-wall landscapes. To a single domain crystal, why the domain-wall landscapes can be a key factor? Especially, the AHE appears in the paramagnetic state above T_C . In summary, the experiments are well done, the data are high quality. But the origins for the observations are not convincing. I suggest the authors should make clear what are the new points compared to the other similar compounds, and clearly answer why this material presents different.

Reply: We are grateful to Reviewer #3 for this insightful comment. In the context of scanning Kerr effect microscope measurements, a previous study proposed the existence of two types of domain boundaries, each characterized by distinct topology and varying interactions with Weyl fermions [Y.

Sun *et al.*, *Phys. Rev. B* **104**, 235119 (2021)]. Although the dimensions of our high-pressure sample are comparable to the size of a single magnetic domain, it is possible that multiple smaller magnetic domains persist within the high-pressure sample. When compared to the larger sample used for ambient-pressure measurements, the relative influence of these magnetic domains in the high-pressure sample may be more pronounced, subsequently exerting a more significant impact on the material's topological properties. It's important to note that in our high-pressure experiments, we did not investigate the temperature-dependent evolution of the anomalous Hall effect in the paramagnetic state at different pressure levels. This is a potential avenue for future research. In an effort to provide a more balanced discussion, we have revised and softened some of the arguments in the discussion section.

We thank Reviewer #3 again, and we hope his/her concerns have been well addressed in our updated manuscript.

List of changes to manuscript

In the main text:

(1) These co-authors, “Pengtao Yang, Bosen Wang, Jin-Guang Cheng, Ulrich Burkhardt, Claudia Felser” have been added.

(2) Page 1, the Abstract:

“, suggesting the great potential of controlling Weyl node positions in CeAlSi” has been revised to “More importantly, we reveal that the Weyl nodes with opposite chirality are moving away from each other upon entering the ferromagnetic phase, which can well explain the enhancement of AHE/ANE.”.

(3) Page 4, the Introduction section:

These sentences “Since the TR symmetry..., further illuminating that the Weyl node positions can be adjusted by magnetism” have been revised to “Since the TR symmetry...The magnetic tunability of the bulk and surface band structure is experimentally realized, which, to the best of our knowledge, has never been reported before in other $RAIPn$ compounds.”.

(4) Page 6:

(a) The first paragraph:

These arguments regarding intrinsic AHE, i.e., “The AHE and ANE have also been theoretically estimated...further analyze the intrinsic AHE [Supplementary Fig. 5(b)].” have been updated.

(b) The second paragraph:

These sentences, “To shed light on the anisotropic nature, we conducted the electrical and thermoelectrical measurements with in-plane magnetic field [see Supplementary Note 4 for more details]. The in-plane transport behavior is more complex due to magnetocrystalline anisotropy, but similar conclusion can be drawn.”, have been added according to the comments raised by Reviewer #2.

(5) Page 8:

(a) These sentences regarding our new ARPES results have been added.

(b) For the ac magnetic susceptibility measurements, these sentences, “To illuminate the pressure evolution...Through a careful comparison of them, we found pressure

may tune the electron correlation effect as well as valence fluctuations (discuss later).” have been added.

(6) Page 10, the Discussion section:

“Under pressure...magnetostriction/magnetoelastic effects” has been revised and updated.

(7) Page 11, the first paragraph:

(a) “Beyond the ferromagnetic transition, it's worth noting that other factors, such as Kondo coupling... For more comprehensive insights, please refer to Supplementary Note 9.” has been added.

(b) The second paragraph:

“At ambient pressure, the enhancement of AHE and ANE across T_c stems...” have been updated.

(c) In the Methods section:

“The as-grown single crystals were characterized by compositional analysis, Laue diffraction pattern, and x-ray diffraction (XRD) measurements [Supplementary Fig. 2]” has been added.

(8) Figures 1, 2 and 3 have been updated, and the captions have been updated accordingly.

In the Supplementary Information:

(1) Supplementary Note 1 has been updated.

(2) Compositional analysis and Laue diffraction pattern have been added in Supplementary Fig. 2.

(3) Calculated anomalous conductivity (AHC) and anomalous Nernst conductivity (ANC) have been plotted in Supplementary Fig. 3.

(4) In-plane anomalous Hall effect and anomalous Nernst effect have been added in Supplementary Note 4.

(5) In Supplementary Note 5, ρ_{yx}^A as a function of ρ_{xx}^2 has been plotted.

(6) Supplementary Note 9 has been updated, and some related discussions have been added.

(7) References have been updated.

REVIEWER COMMENTS

Reviewer #1 (Remarks to the Author):

The manuscript “Tunable Weyl nodes via magnetism and pressure in the ferromagnetic Weyl semimetal CeAlSi” has been significantly revised, which also involved inclusion of new results. Notable among these are new ARPES measurements showing that there is a difference in Weyl nodes positions between FM and PM phases. This look interesting, although the effect is rather small compared the noise level. However, not all issues have been resolved and some new ones have emerged.

Firstly, I would like to thank the authors for expressing their gratitude for my earlier comments. I have some additional remarks on their responses as well as the new content:

- Figs. 8a and 8c have been added to support the claim that the hydrostatic pressure tunes the electron-electron correlations. The latter presents the pressure dependence of the parameters A and n from the resistivity equation: $r = r_0 + AT^n$. It is shown that for the entire pressure range $n \sim 2$, whereas A peaks at $p \sim 10$ GPa. Unfortunately, we cannot see how the data are actually fitted. This is particularly intriguing for $p \sim 16$ GPa, where the slope of $r(T)$ is negative at low temperature (Fig. S8b), but A from Fig. S8c appears to be positive (and $n \sim 2$). In addition, the similarity between the pressure evolution of A and the magnetic transition under pressure is said to reflect that pressure not only tune the magnetic transition but also the electron-electron correlations. Much simpler explanation may be the effect of charge carriers scattering on ferromagnetic fluctuations near the transition.

- Calculations presented in Fig S3d indicate that anomalous σ_{yz} and σ_{zx} should be almost zero, which seems to contradict the results from Fig. S4. It is shown there that ρ^A_{zy} has a similar magnitude to ρ^A_{yx} and it is even said that the results closely mirror what has been observed when a magnetic field is applied along the c-axis. The anomalous Nernst effect for B || a-axis is also sizeable. How can this discrepancy be explained?

- The fit of r^A_{xy} to the Tian-Ye-Jin (TYJ) model shown in Fig. S5b is interpreted as a sign of an intrinsic behaviour below T_c , where r^A_{yx} is roughly proportional to r^2_{xx} . On the contrary, for $T > T_c$ the deviation from “linearity” is suggested to originate from extrinsic contributions. In such a case, should not r^A_{yx} be proportional to r_{xx} ? Furthermore, the data for sample S1 shown in Fig. S5a are indicated at the same time as a verification of the intrinsic nature of the AHE in CeAlSi. Are they limited to $T < T_c$?

- If the authors insist on stating in the abstract that CeAlSi offers a range of potential applications in spintronics and thermoelectrics, in my opinion they should provide a more detailed explanation of these in the text. It was written that manipulating the positions of Weyl node results in an enhancement of the Anomalous Nernst Effect, but over the entire temperature range the Nernst signal only reaches about 0.8 $\mu\text{V}/\text{K}$ at 15 T.

- I think that the claim that the interplay between magnetism and topology remains mysterious but is expected to be promising for the realization of novel topological states requires further explanation.

To summarize, in my opinion the ambiguities in the current version of the manuscript do not allow for a positive recommendation.

Reviewer #2 (Remarks to the Author):

The authors have addressed my comments and revised the paper. It can be accepted for publication.

Reviewer #3 (Remarks to the Author):

The authors have clarified all of my concerns and made significant modification. I recommend it for acceptance for publication in its revised version.

Reviewer #1 (Remarks to the Author):

The manuscript “Tunable Weyl nodes via magnetism and pressure in the ferromagnetic Weyl semimetal CeAlSi” has been significantly revised, which also involved inclusion of new results. Notable among these are new ARPES measurements showing that there is a difference in Weyl nodes positions between FM and PM phases. This look interesting, although the effect is rather small compared the noise level. However, not all issues have been resolved and some new ones have emerged.

Reply: We extend our sincere appreciation to Reviewer #1 for re-evaluating our manuscript and providing exceptionally insightful comments. It is our intention to meticulously address each of his/her comments or concerns individually.

Firstly, I would like to thank the authors for expressing their gratitude for my earlier comments. I have some additional remarks on their responses as well as the new content: - Figs. S8a and S8c have been added to support the claim that the hydrostatic pressure tunes the electron-electron correlations. The latter presents the pressure dependence of the parameters A and n from the resistivity equation: $\rho = \rho_0 + AT^n$. It is shown that for the entire pressure range $n \sim 2$, whereas A peaks at $P \sim 10$ GPa. Unfortunately, we cannot see how the data are actually fitted. This is particularly intriguing for $P \sim 16$ GPa, where the slope of $\rho(T)$ is negative at low temperature (Fig. S8b), but A from Fig. S8c appears to be positive (and $n \sim 2$). In addition, the similarity between the pressure evolution of A and the magnetic transition under pressure is said to reflect that pressure not only tune the magnetic transition but also the electron-electron correlations. Much simpler explanation may be the effect of charge carriers scattering on ferromagnetic fluctuations near the transition.

Reply: We thank Reviewer #1 for highlighting this aspect. As displayed in Fig. R1, we fit the low-pressure data within the magnetic ordering state, whereas the fitting encompassed data below 100 K for higher pressures, attributed to the profiles of typical metallic behavior in resistivity. For pressures at 15.9, 17.8 and 19.8 GPa, due to the resistivity's divergence at low temperature, we did not fit the data. In Figs. S8(c), the fitting parameters do not contain the data at 15.9, 17.8 and 19.8 GPa. To enhance clarity, Fig. R1 has been incorporated into Fig. S8 within the Supplementary Information.

We concur with Reviewer #1 regarding the potential impact of charge carrier scattering on ferromagnetic fluctuations proximate to the transition, which may elucidate the pressure evolution of A and n . However, the pressure-dependent A values (compared with typical heavy-fermion compounds) is not solely attributable to the scattering mechanism, and we contend that electron-electron correlations, to some extent, are also tuned by pressure. To ensure rigor, certain discussions have been revised in response to the insights provided by Reviewer #1.

Figure R1. Longitudinal resistivity at various pressures. The red solid lines depicted in **a** and **b** represent the fitting achieved through a power law, namely, $\rho = \rho_0 + AT^n$. In left panel, the data was fitted within the magnetic ordering state, whereas in right panel, the fitting encompassed data below 100 K, attributed to the display of typical metallic behavior in resistivity. Notably, data points at 15.9, 17.8 and 19.8 GPa were omitted from the fitting process due to the resistivity's divergence at low temperatures.

- Calculations presented in Fig S3d indicate that anomalous σ_{yz} and σ_{zx} should be almost zero, which seems to contradict the results from Fig. S4. It is shown there that ρ_{zy}^A has a similar magnitude to ρ_{yx}^A and it is even said that the results closely mirror what has been observed when a magnetic field is applied along the c -axis. The anomalous Nernst effect for $B // a$ -axis is also sizeable. How can this discrepancy be explained?

Reply: We express our gratitude to Reviewer #1 for highlighting this aspect. To accurately calculate the anomalous Hall conductivity (AHC), achieving convergence demands an extensive number of k -points. Typically, millions of k -points are indispensable to ensure precise outcomes [X. Wang *et al.* *Phys. Rev. B* **74**, 195118 (2006); G.-Y. Guo and T.-C. Wang, *Phys. Rev. B* **96**, 224415 (2017)]. Consequently, our previous AHC calculation using a $50 \times 50 \times 50$ Brillouin Zone (BZ) mesh might yield significantly inaccurate values. In this context, we conducted a new AHC computation employing an enlarged $200 \times 200 \times 200$ Γ -centered k mesh, which is now updated in Supplementary Fig. 3(d). This revised result aligns closely with findings reported by another research group [M. S. Alam *et al.* *Phys. Rev. B* **107**, 085102 (2023)], suggesting that the discrepancy could be attributed to the utilization of an inadequately dense k mesh.

- The fit of ρ_{yx}^A to the Tian-Ye-Jin (TYJ) model shown in Fig. S5b is interpreted as a sign of an intrinsic behaviour below T_c , where ρ_{yx}^A is roughly proportional to ρ_{xx}^2 . On the contrary, for $T > T_c$ the deviation from “linearity” is suggested to originate from extrinsic contributions. In such a case, should not ρ_{yx}^A be proportional to ρ_{xx} ? Furthermore, the data for sample S1 shown in Fig. S5a are indicated at the same time as a verification of the

intrinsic nature of the AHE in CeAlSi. Are they limited to $T < T_c$?

Reply: We extend our gratitude to Reviewer #1 for this insightful comment. Figure R2(a) shows the ρ_{yx}^A as a function of ρ_{xx} plot. Observably, the data above T_c (specifically beyond 15 K where magnetic fluctuations become prominent) does not exhibit a linear dependence.

In Fig. S5a, we plot the data below 15 K. We have re-plotted the $|\sigma_{xy}^A|$ as a function of σ_{xx} profile, now incorporating higher-temperature data, shown in Fig. R2(b). For anomalous Hall effect, there are three broad regimes have been identified [N. Nagaosa *et al.*, *Rev. Mod. Phys.* **82**, 1539 (2010)]: (i) a high conductivity regime ($\sigma_{xx} > 10^6 \Omega^{-1} \text{cm}^{-1}$) in which a linear contribution to $\sigma_{yx}^A \sim \sigma_{xx}$ due to skew scattering; (ii) an intrinsic or scattering-independent regime in which σ_{yx}^A is roughly independent of σ_{xx} [$10^4 \Omega^{-1} \text{cm}^{-1} < \sigma_{xx} < 10^6 \Omega^{-1} \text{cm}^{-1}$]; (iii) a bad-metal regime [$\sigma_{xx} < 10^4 \Omega^{-1} \text{cm}^{-1}$] in which σ_{yx}^A decreases with decreasing σ_{xx} at a rate faster than linear. As evidenced above 15 K, where magnetic fluctuations begin to exert dominance [as illustrated in Figs. 1(f) and 1(k)], the system appears to reside within the bad-metal regime. To address this comment, Fig. S5a has been updated, accompanied by additional discussions for clarification.

Figure R2. (a) ρ_{yx}^A as a function of ρ_{xx} plot for S1 at ambient pressure. (b) $|\sigma_{xy}^A|$ as a function of longitudinal conductivity σ_{xx} .

- If the authors insist on stating in the abstract that CeAlSi offers a range of potential applications in spintronics and thermoelectrics, in my opinion they should provide a more detailed explanation of these in the text. It was written that manipulating the positions of Weyl node results in an enhancement of the Anomalous Nernst Effect, but over the entire temperature range the Nernst signal only reaches about 0.8 $\mu\text{V}/\text{K}$ at 15 T.

Reply: We are thankful to Reviewer #1 for bringing this to our attention. In response to this

comment, we have revised certain arguments in the abstract section to enhance clarity and accuracy.

These sentences, “These findings indicate that CeAlSi provides a unique and tunable platform for exploring exotic topological physics and electron correlations, as well as catering to an array of potential applications, such as spintronics and thermoelectrics.” have been revised to “These findings indicate that CeAlSi provides a unique and tunable platform for exploring exotic topological physics and electron correlations, as well as catering to potential applications, such as spintronics.”.

- I think that the claim that the interplay between magnetism and topology remains mysterious but is expected to be promising for the realization of novel topological states requires further explanation. To summarize, in my opinion the ambiguities in the current version of the manuscript do not allow for a positive recommendation.

Reply: We sincerely appreciate Reviewer #1 for his/her valuable comment. In magnetic systems, the impact of magnetic structures and the reorientation of magnetic moments, controlled by external parameters such as magnetic field or pressure, leads to diverse effects on emerging topological states and properties, which varies significantly from one system to another. Recently, this topic is of great interest in topological physics. For example, in NdAlSi system, Weyl-mediated helical magnetism has been reported [J. Gaudet *et al.*, *Nat. Mater.* **20**, 1650–1656 (2021)]. Quite recently, a paper has surfaced reporting the emergence of Weyl fermions due to ferrimagnetism in NdAlSi as well. [C. Li *et al.*, *Nat. Commun.* **14**, 7185(2023)]. As mentioned before, for PrAlSi and SmAlSi, our previous work demonstrated a negligible effect of the magnetism on their electronic structures [R. Lou *et al.*, *Phy. Rev. B* **107**, 035158 (2023)]. For CeAlSi, magnetism has an entirely different impact on electronic structures and topological properties, which makes it a unique platform to study the interplay between topology and magnetism. To avoid ambiguities, this sentence, “The interplay between magnetism and topology remains mysterious but is expected to be promising for the realization of novel topological states.”, has been revised to “The intricate relationship between magnetism and topology continues to be complex, yet holds promise for unlocking new and exotic topological states.”.

We express our gratitude once more to Reviewer #1 and sincerely hope that our response adequately addresses his/her concerns.

REVIEWERS' COMMENTS

Reviewer #1 (Remarks to the Author):

The authors have addressed the issues raised and, in my opinion, the manuscript can be accepted for publication.